# A dynamic informed deep learning method for future estimation of laboratory stick-slip

Enjiang Yue[1,2], Mengjiao Qin[3], Linshu Hu[1,2], Bryan Riel[1], Sensen Wu[1,2], and Zhenhong Du[1,2*]

[1]School of Earth Sciences, Zhejiang University, Hangzhou 310058, China.
[2]Zhejiang Provincial Key Laboratory of Geographic Information Science, Hangzhou 310058, China.
[3]School of Safety Science and Emergency Management, Wuhan University of Technology, Wuhan, 430081, China.

*Corresponding author*: Zhenhong Du (duzhenhong@zju.edu.cn)

**Abstract.** Fault activity modelling is vital for earthquake monitoring, risk management, and early warning. Studies on laboratory earthquakes are instrumental for modelling natural fault ruptures and enhancing our understanding of natural earthquake dynamics. Recently, machine learning methods have proven effective in predicting instantaneous fault stress in laboratory settings and fault activities on Earth. However, these methods have struggled to obtain steady future predictions because of the lack of understanding of the complex dynamics of highly nonlinear laboratory fault slip systems. To address this, we introduce the Hankel–Koopman autoencoder (HKAE), a novel method inspired by dynamic system theories. The HKAE performs dynamic modelling of laboratory fault systems and provides a continuous estimation of the future state of the system. It has been used in experiments with different slip behaviours and has the ability to predict shear stress variation during a slip cycle and slip activity during long-term seismic cycles. The HKAE outperforms traditional statistical methods while achieving results comparable to cutting-edge deep learning methods across multiple prediction scales. This is particularly evident in its accurate prediction of the stress release phase and precise estimation of the slip interval. More importantly, through dynamic theory and operator analysis in latent space, the HKAE provides insights into the stability of laboratory slip systems rather than full end-to-end black-box predictions. The ability of the HKAE to decompose, model and reveal complex temporal dynamics highlights its potential in monitoring of sparsely observed geophysical systems with cyclic characteristics, such as natural faults.

## 1 Introduction

Modelling fault activity is crucial for understanding patterns of seismic activity, monitoring and predicting earthquakes, and estimating seismic hazards. Laboratory earthquake studies have contributed to modelling natural fault ruptures and enhancing our understanding of natural earthquakes (Johnson et al., 2021). These studies indicate a similar mechanism between slow and fast slip (Hulbert et al., 2019) and aid in extracting physical property changes in faults from dense earthquake records (Rouet-Leduc et al., 2019). Machine learning has proven effective in extracting information about the rupture behaviour of laboratory earthquakes from acoustic emission signals for instantaneous prediction. Rouet-Leduc et al. (2017) reported that the random forest method can be used to accurately predict the time-to-failure via acoustic emissions. Subsequently, stress variation, which

is a crucial physical feature of faults, has been identified and evaluated from acoustic emissions via XGBoost, enabling further analysis of the acoustic signals (Rouet-Leduc et al., 2018). Lubbers et al. (2018) reported that the event catalogue, which is more available for natural earthquakes, can also be used to predict the transient fault mechanism during laboratory earthquakes. Active-source seismic data are also valid data sources for predicting instantaneous fault behaviour (Shokouhi et al., 2021).

Jasperson et al. (2019) and Karimpouli et al. (2023) discussed prediction methods, such as traditional machine learning methods, neural networks and explainable machine learning methods. An assessment of the transferability across diverse experiments and simulations was conducted, highlighting the critical role of applying laboratory methods to in-field models (Wang et al., 2021; Borate et al., 2023).

While most studies have focused on instantaneous predictions, several have explored future predictions. The state-of-the-art

sequence modelling architecture, namely, the transformer, has shown promise in extracting information for predicting friction in the future from continuous acoustic emission signals (Wang et al., 2022). The model's attention score reveals that the closer the fault is to the rupture moment, on the basis of the friction data, the stronger the stress drop in seismic records. Laurenti et al. (2022) reported that laboratory fault zone stress can be autoregressively inferred. Additionally, spatial dimensions have been introduced for autoregressively predicting surface velocity fields during laboratory fault slips (Mastella et al., 2022).

Although these studies underscore the potential for inferring the future behaviour of fault slips, they face challenges in modelling stability and predicting future behaviour owing to the complex dynamics of laboratory fault slip systems. Gualandi et al. (2023) proposed that earthquake cycles in laboratory experiments can be characterized as systems with average dimensions similar to those of natural earthquakes. The Lyapunov exponent analysis reveals the predictability within a certain period, albeit with deterministic and stochastic chaotic behaviours, which are challenging to model via machine learning

methods designed from traditional statistical knowledge.

Physics-informed machine learning methods constitute a framework for geoscientific applications (Degen et al., 2023) such as glacier modelling (Riel et al., 2021), ocean modelling (Hammoud et al., 2022) and solid-Earth modelling (Okazaki et al., 2022). These methods introduce prior domain knowledge, which is the key factor in geoscientific analysis, while leveraging the benefits of machine learning. Recent advancements in dynamic theory, on the basis of Koopman theory (Koopman, 1931),

have shown efficacy in integrating dynamic insights within a data-driven framework, yielding results that are more aligned with dynamic situations (Karniadakis et al., 2021). Various methods based on the Koopman theory have been acknowledged as powerful for modelling and deciphering complex nonlinear dynamic systems (Brunton et al., 2022), such as fluid mechanics (Brunton et al., 2020), and have found applications in geophysical fields, including climate (Li et al., 2020; Froyland et al., 2021), ocean variability (Franzke et al., 2022), and electromagnetic fields (Brunton et al., 2017; Lintner et al., 2023).

Given the complicated dynamics of laboratory fault slip systems, we propose a deep learning method that involves the Koopman theory. Instead of defining the problem as a statistical time series forecast task, we take the shear stress time series as one of the observations in a laboratory slip system and infer changes in the future state of the system through methods inspired by dynamic systems theory. Generally, we reconstruct the phase space of the system via delayed embedding theory and linearize its complex dynamics via the Koopman theory to perform future inference and further dynamic analysis.

Laboratory fault systems with different slip behaviours under different prediction horizons are adapted to evaluate the effectiveness of Hankel–Koopman autoencoder (HKAE) modelling.

## 2 Materials, methods and models

### 2.1 Laboratory stick–slip data

Our study incorporates two categories of data: data drawn from laboratory experiments carried out with biaxial shear equipment
and data derived via numerical simulations. The experimental data from the biaxial shear equipment were obtained from Chris Marone's laboratory (Laurenti et al., 2022). Different shear materials are situated between the two plates to which positive pressure and shear force are exerted from each side, and the equipment is used to record the mechanistic changes (Supplementary Information. Table S1) in the system during the shear process (Figure 1a-b). This results in time series data recorded at a temporal sampling rate of 0.001 s. Here, we focus mainly on the variation in shear stress because of its direct
indication of the onset of fault slip in the laboratory. Experiments 4581 (Exp. 4581) and 5198 (Exp. 5198) demonstrate cyclic slow and fast slip behaviours, respectively, whereas Experiment 4679 (Exp. 4679) involves a switch between two types of slip behaviours (Figure 1c). We derive numerical simulation data from the model (Figure 1d) in Gualandi's work (2023). This model is based on the rate-state-friction law (Dieterich et al., 1979), and we set the initial normal stress to $\tau_{n0} = 17.003\ Mpa$ and the initial state vector to $[x_0, y_0, z_0, u_0] = [0.05, 0.0, 0.0, 0.0]$ to generate fast-slow-switching slips as examples in the
simulation.

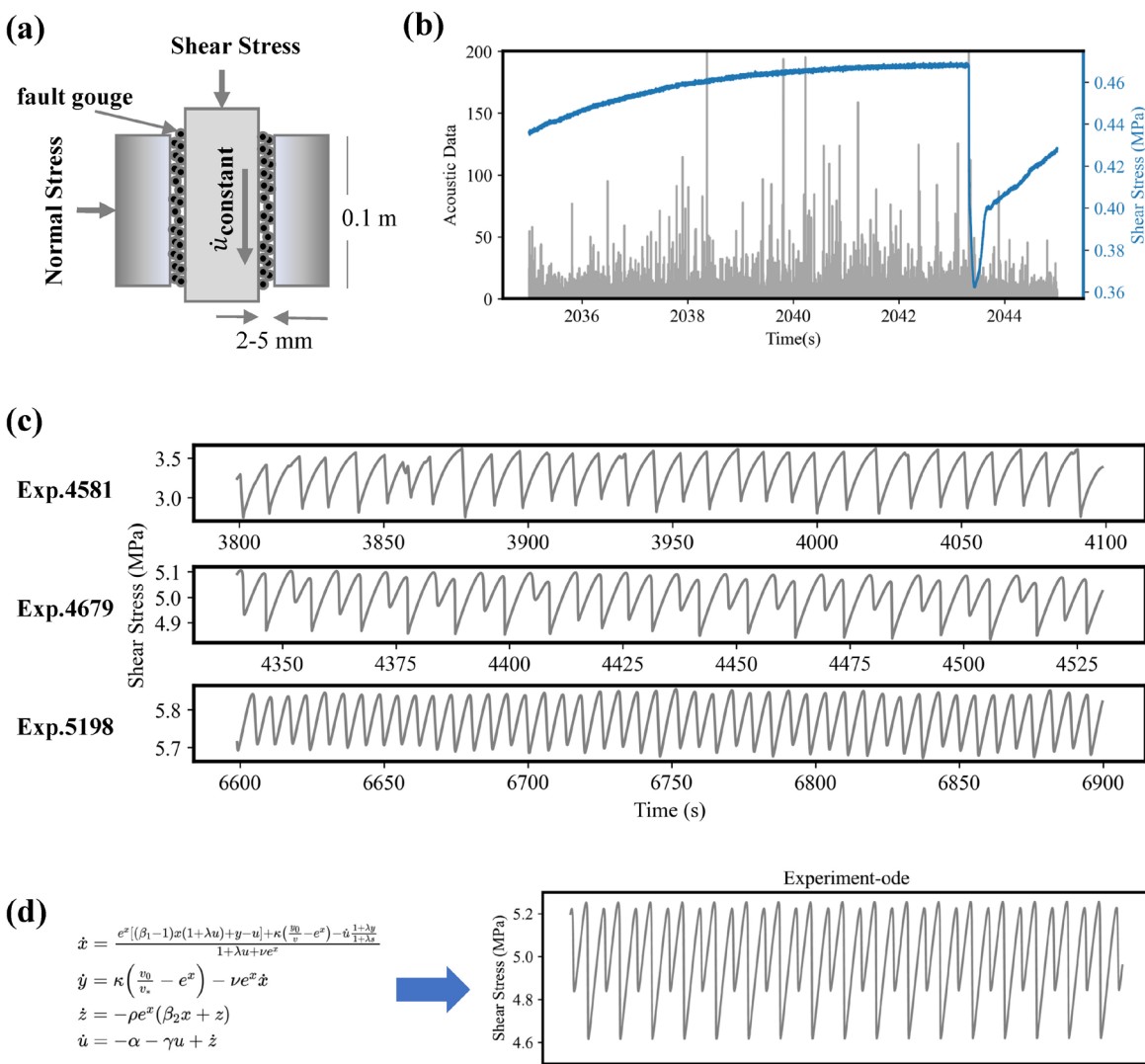

Figure 1: (a) Laboratory fault slip experimental settings and (b) recorded data. Acoustic emission data are recorded in grey, and shear stress time series are recorded in blue. (c) Three modelling experiments with different slip behaviours. (d) Simulated shear stress, where $[x, y, z, u] = [ln\left(\frac{v}{v_*}\right), \frac{\tau_f - \tau_0}{a\sigma_{n0}}, \frac{1}{\lambda\beta\sigma_{n0}}(\phi - \phi_0), -\frac{1}{\lambda}\frac{p}{\sigma_{n0}}]$ represents the system state.


## 2.2 Dynamic system theories

### 2.2.1 Koopman theory

Laboratory earthquakes can be conceptualized as being governed by a dynamic system, and shear stress can be regarded as a measurement within this system and is described by Eq. (1):

$$s_{t+1} = F(s_t) \tag{1}$$

where $s_t$ is the state of the laboratory slip system and where $F$ represents the governing function of the system. The shear stress $x_t$ can be viewed as an observation of the laboratory slip system state $s_t$.

Koopman theory is a mathematical theoretical framework. All finite-dimensional nonlinear systems can evolve in an alternative space through the mapping function $g$ and the infinite-dimensional Koopman operator $K$ (Koopman, 1931; Brunton et al., 2021). The Koopman operator in the transformed space can be used directly to perform the linear evolution of the system state, as shown in Eqs. (2)-(3) and Figure 2.

$$Kg(x_t) = g\big(F(x_t)\big) = g(x_{t+1}) \tag{2}$$

$$x_{t+1} = g^{-1}Kg(x_t) \tag{3}$$

Owing to the linear properties of the Koopman operator, linear methods such as spectral decomposition can be employed in the operator for enhanced analysis, prediction, and control. The dimension of the learned approximate Koopman operator indicates the dynamic modes needed to describe the dynamic process, which can be decomposed as follows:

$$K = V\Lambda V^{-1} \tag{4}$$

where $V = [v_1, v_2, \dots, v_k]$ are the eigenvectors of $K$ and where $\Lambda = [\lambda_1, \lambda_2, \dots, \lambda_k]$ are the eigenvalues of $K$. Each eigenvalue describes the strength and oscillatory properties of its corresponding dynamic component:

$$b_j^k = b_0^k e^{\frac{j}{\Delta t}\log\lambda_i} \tag{5}$$

Here, $\boldsymbol{b^k} = [b_1^k, b_2^k, \dots, b_N^k]$ represents the temporal evolution of the $k^{th}$ dynamic mode, and $j$ represents the time step.

Provided that we know the current state, we can infer the system's future behaviour incrementally via the mapping function $g$ and the linear operator $K$, and the dynamic characteristic can be explored through the eigen decomposition of $K$, for example, to explore the main components driving the evolution of dynamic systems (Brunton et al., 2021), the pattern of its growth (Schimid et al., 2010), etc.

The most important advantage of Koopman theory is that it linearizes the dynamics from complex laboratory slip systems and then supports the analysis of system behaviour via linear analysis tools (e.g., singular value decomposition), which offers insights from the perspective of dynamic systems rather than simply statistical inference.

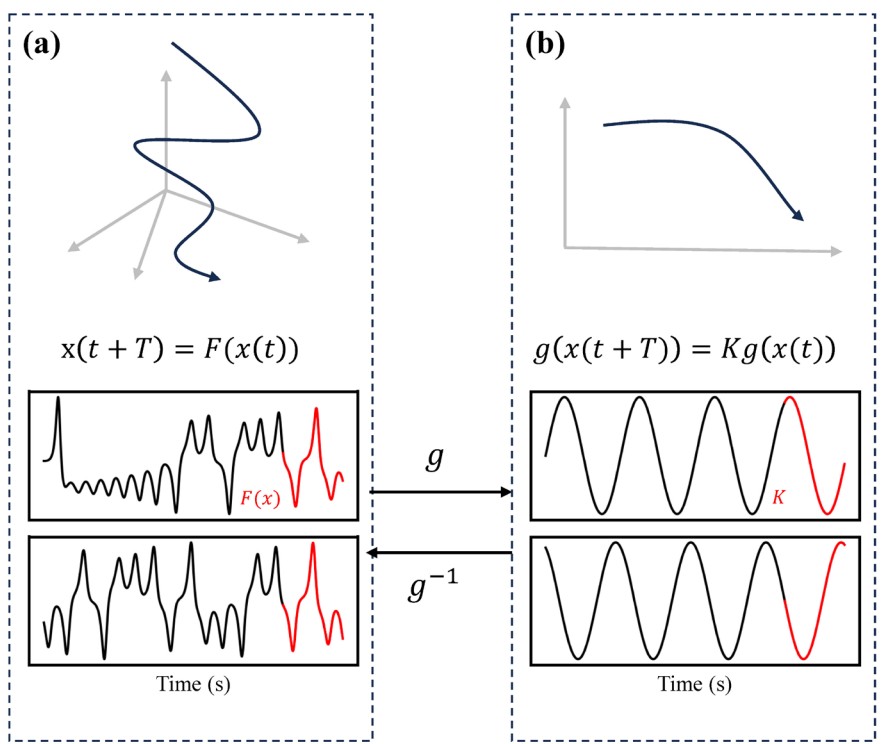

**Figure 2: Conceptual illustration of the transformation between the nonlinear trajectory of the high-dimensional state, with F (a) and a linear operator K representing dynamics (b), taking the Lorenz63 system as an example. Lorenz63 system observations are used to represent the transformation (Lorenz, 1963).**

### 2.2.2 Delay embedding theory

For analysing dynamic systems via Koopman theory, it would be useful to have direct system states or observables that carry information about the main changes in the system. However, in the real world, we often obtain state quantities with a limited signal-to-noise ratio or observations that do not carry all the information about the changes in the system, i.e., partial observations. Inferring the state change of a laboratory slip system from shear stress can be viewed as inferring the future evolution of the system from very limited observations (Arbabi et al., 2017). Here, we introduce delay embedding theory to reconstruct the system behaviour. Delay embedding theory supposes that topological reconstruction of attractors from the original high-dimensional system, also known as phase space reconstruction, can be performed using only the observed univariate long time series (Takens, 1981). We define $h$ as the embedded variable, taking $H = [\mathbf{h}_1, \mathbf{h}_2, ..., \mathbf{h}_i]$ as the input. The embedding process is described in Eq. (6) with the parameters, the embedded dimension $d$ and the delay time $\tau$. The delay time $\tau$ is usually 1 in most situations (Brunton et al., 2017).

$$\mathbf{H} = \begin{bmatrix} x_1 & x_2 & \cdots & x_i \\ x_{1+\tau} & x_{2+\tau} & \cdots & x_{i+\tau} \\ \vdots & \vdots & \ddots & \vdots \\ x_{1+(d-1)\tau} & x_{2+(d-1)\tau} & \cdots & x_{i+(d-1)\tau} \end{bmatrix} = \begin{bmatrix} x_1 & x_2 & \cdots & x_i \\ x_2 & x_3 & \cdots & x_{i+\tau} \\ \vdots & \vdots & \ddots & \vdots \\ x_d & x_{d+1} & \cdots & x_{i+(d-1)} \end{bmatrix} = [\mathbf{h}_1, \mathbf{h}_2, \ldots, \mathbf{h}_i] \tag{6}$$

Here, we aim to utilize historical shear stress observations to ascertain the evolution of future states or to discern the relationship between historical shear stress $(x_{t-K}, x_{t-K+1}, \ldots, x_t)$ and future states $(x_{t+1}, \ldots, x_{t+L-1}, x_{t+L})$. On the basis of delay embedding, the relationship between $[\mathbf{h}_1, \mathbf{h}_2, \ldots, \mathbf{h}_i]$ and $[\mathbf{h}_{i+1}, \mathbf{h}_{i+2}, \ldots, \mathbf{h}_{i+L}]$ can be deconstructed to determine the

135 mapping function $g$ and operator $K$.

Figure 3 shows the process of delayed embedding. To verify the retention of the original system's topological relations in the phase space following delayed embedding, we often resort to singular value decomposition to identify the system's three primary components and use the corresponding singular vectors to open the space to represent the evolution of the system. Taking the Lorenz system as an example (Lorenz, 1963), Figure 3a represents the tensor space using the state variables of the

140 original system. The observations yield only a single scalar series, as depicted in Figure 3c. Figure 3b shows the first three singular vectors that result from delayed embedding, which is diffeomorphic with the system represented by Figure 3a, i.e., they are considered to represent the same dynamic system behaviour. This approach has been shown to be effective in enhancing the feature dimensions from the data, and it has also been shown to be important in governing equation extraction (Bakarji et al., 2023) and modelling with dynamic mode decomposition (DMD) (Avila and Mezic, 2020). In summary, we

utilize delay embedding theory to reconstruct a laboratory slip system represented by a single shear stress variable, thereby supporting subsequent Koopman mode decomposition and prediction.

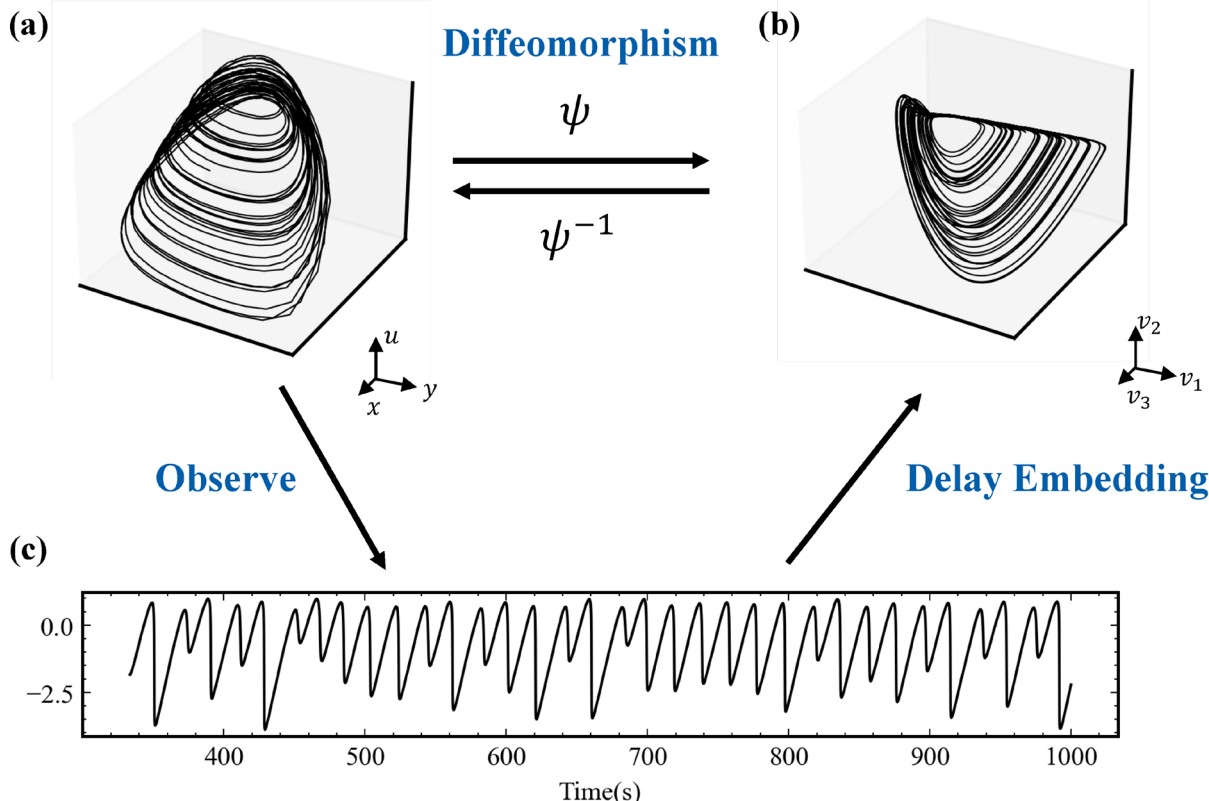

**Figure 3: Conceptual process of delay embedding theory. (a) Representation of laboratory system behaviour using the original state $[u, x, y]$. (b) Representation of laboratory slip system behaviour via singular vectors $[v_1, v_2, v_3]$ from the SVD result of delay embedded observations $x$. (c) Single observation of the laboratory slip system and shear stress used in this system. The data are simulated on the basis of Gualandi's model (2023).**

## 2.3 Architecture of the Hankel–Koopman autoencoder (HKAE)

Here, we propose the Hankel–Koopman autoencoder (HKAE), which synthesizes the inductive bias of dynamic system theories and the nonlinear fitting ability of deep learning. This model encompasses three key modules (Figure 4):

(1) For the delay embedding module, by employing delay embedding theory, the shear stress time series $(x_{t-d}, x_{t-d+1}, \ldots, x_t)$ are reconstructed in phase space to obtain their Hankel matrix $H = [\mathbf{h}_1, \mathbf{h}_2, \ldots, \mathbf{h}_i]$ in this module.

(2) For the mapping learning module, the powerful nonlinear fitting capabilities of deep learning have been demonstrated to effectively learn the mapping, thereby approximating an optimal Koopman operator (Takeishi et al., 2017; Lusch et al., 2018; Azencot et al., 2020). Here, we utilize an encoder–decoder backbone incorporating a 3-layer multilayer perceptron (MLP) to learn the mapping between the phase space and Koopman invariant subspaces.

(3) For the Koopman evolution module, in this module, the Koopman operator is represented as a layer of neurons consisting solely of weights and devoid of bias. Following the encoding process, the system is mapped into Koopman invariant subspaces, where the Koopman operator is applied to facilitate system evolution. Multistep prediction is achieved through the iterative application of the same set of operators corresponding to the predefined prediction steps. The decoded evolution results remain in phase space. A re-embedding process is subsequently applied to derive the predicted shear stress, which is implemented by selecting the final value of each evolved result.

The loss of this model includes 2 parts, namely, reconstruction and evolution loss, as shown in Eqs. (7)-(9). The reconstruction loss is set to minimize the loss that occurred during the mapping process, whereas the evolution loss is set to minimize the loss of linear evolution achieved by the Koopman operator.

$$\varepsilon_{reconstruct} = \frac{1}{2L} \sum_{i=1}^{L} \parallel h_i - \widehat{h}_i \parallel_2^2 \tag{7}$$

$$\varepsilon_{evolution} = \frac{1}{2KL} \sum_{j=1}^{K} \sum_{i=1}^{L} \parallel h_{i+j} - \widetilde{h}_{i+j} \parallel_2^2 \tag{8}$$

$$Loss = \varepsilon_{reconstruct} + \lambda \varepsilon_{evolution} \tag{9}$$

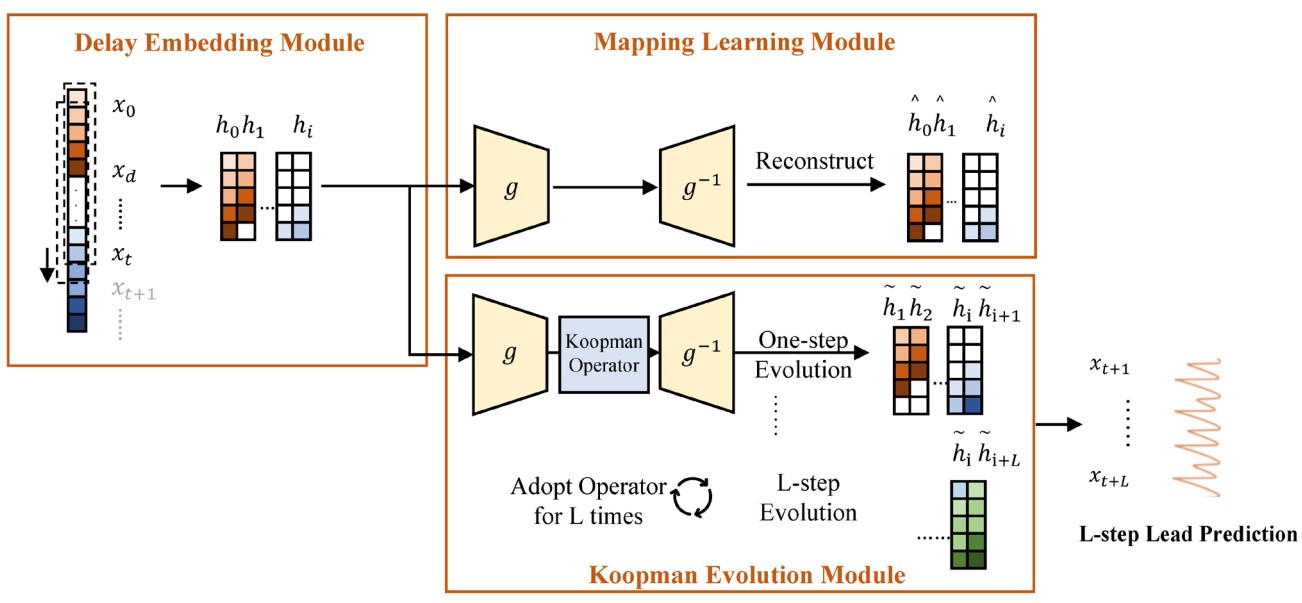

**Figure 4: Architecture of the HKAE.**

## 2.4 Time series prediction models from a statistical perspective

### 2.4.1 Long short-term memory (LSTM)

Long short-term memory (LSTM) is a widely used deep learning model for sequence modelling that addresses the challenge of vanishing and exploding gradients in long sequence data (Hochreiter and Schmidhuber, 1997). It has become a powerful tool in temporal modelling in the Earth sciences (Feng et al., 2023). The core of the LSTM is the LSTM cell. An LSTM cell consists of a cell state and a set of gates (input, forget and output gates). The input gate determines the new input information to add to the cell state, whereas the forget gate chooses the information to drop in the cell state. The output gate controls how

the cell state is mapped to the output. The structure of an LSTM cell is shown in Figure 5a.

### 2.4.2 Temporal convolutional network (TCN)

A temporal convolutional network (TCN) is a sequence modelling method inspired by convolutional operations that are widely used in image processing (Bai et al., 2018). It has been regarded as a state-of-the-art model in laboratory fault slip modelling (Laurenti et al., 2022). The core of the TCN is a series of sequence convolution and pooling blocks. In general, the features of

the input sequence are extracted through dilated convolution with a sliding window. The pooling layer is adapted for dimensional reduction. The weight norm and drop out are used to improve the robustness of the model. The structures of the TCN model and TCN block are shown in Figure 5b.

### 2.4.3 Multilayer perceptron (MLP)

In addition to the two-sequence modelling method, we also test the simple MLP model. Here, we take the historical sequence as the input feature and predict the sequence as the output, as shown in Figure 5. In this way, the MLP is equivalent to performing a 1*1 convolution directly in the time dimension, which is equivalent to a TCN model with the dilation set to 1.

**(a) Long Short-Term Memory (LSTM) Architecture**

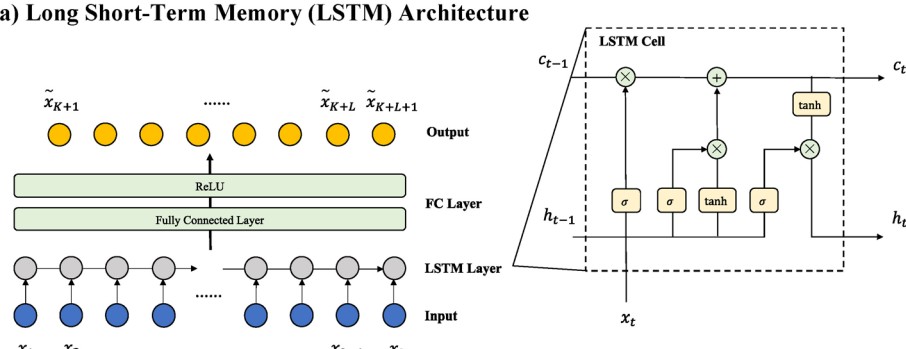

**(b) Temporal Convolutional Network (TCN) Architecture**

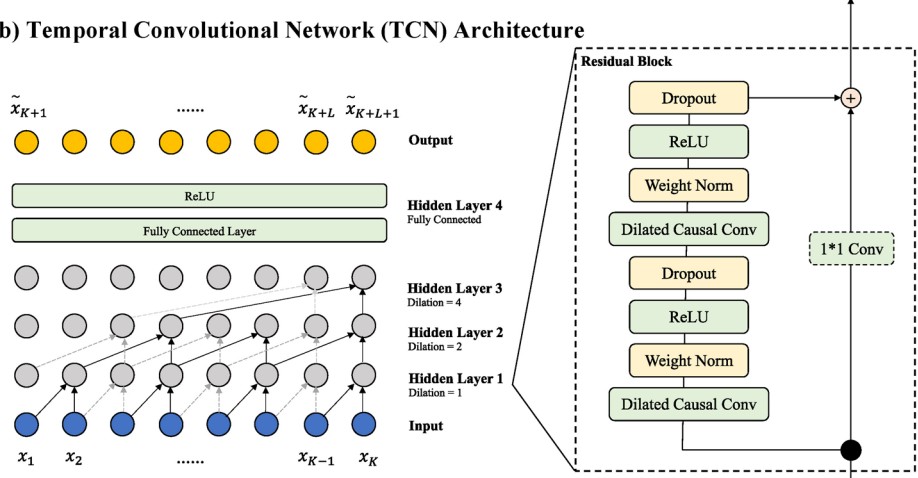

**(c) Multi-layer Perceptron (MLP) Architecture**

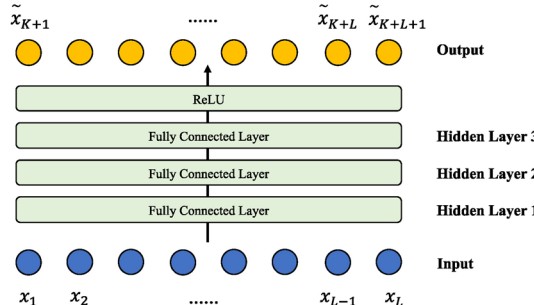

**Figure 5: Architecture of comparative deep learning models**


## 2.5 Experimental settings and evaluation

In our experiments, we utilized three datasets with distinct slip features and one numerical simulation result to assess the robustness of our model. Numerical simulation data are primarily used to illustrate the dynamic modelling process (Figure 7) because of their relatively simple system trajectories.

We employed a unified time sampling rate of 0.1 s. Percentages of 50%, 10%, and 40% were applied to each dataset for training, validation, and testing, respectively. The sliding window method was employed to generate data sequences for model training and inference. For the input dataset $[x_1, x_2, \ldots, x_N]$, we sample the historical $K$ steps to predict the future $L$ steps. The prediction of the sliding window can be presented as follows:

$$(x_i, \ldots, x_{i+K}) \rightarrow (\tilde{x}_{i+K+1}, \ldots, \tilde{x}_{i+K+L}) \tag{10}$$

where $i = (1, \ldots, N - K - L + 1)$ represents the number of times the window slides.

The hyperparameters of the HKAE model were divided into three main categories corresponding to the three model modules. The first set comprises the embedding dimension and delay time in the delay embedding module. For the delay time, we took a value of 1, on the one hand, to construct the Hankel matrix to satisfy the prediction requirements, and on the other hand, Brunton et al. (2017) reported that, in practice, most of the systems have a delay time of 1. With a delay time setting of 1, the

embedding dimensions in our model architecture are numerically equal to the number of steps in which the historical information is used. Given the experimental setup in Laurenti's work (2022), we suggest that sufficient historical information is needed to predict future changes. To evaluate both intra- and intercycle scenarios of slip, we constructed an experimental setup using 10 s of historical data to predict 3 s in the future and 20 s of historical data to predict 10 s in the future (Figure S1 in the Supplementary Information), corresponding to embedded dimensions of 200 and 100. Considering that the slip system

experiences an instantaneous increase in system dimension during the sliding process, we retain a higher embedding dimension setting (Gualandi et al., 2023). For comparison, we used consistent parameter settings across different slip data.

The number of MLP layers in the mapping learning module was set to 3 on the basis of previous works and results. Finally, the dimension of the Koopman operator module was set to 10 on the basis of the data and performance. We adopted a batch size of 64, L2 loss as the loss function, and Adam as the optimization algorithm. Weight decay and gradient crip skills were

adopted to improve performance. Given the single-step evolutionary nature of the Koopman operator, to keep it robust for future leading predictions by learning the dynamics, we used a multistep trick for error estimation during model training (Eq. (8)).

For the comparative models, LSTM and MLP obey similar settings in Laurenti's work. The detailed key parameter settings are listed in Table 1. To achieve multistep prediction, we add a fully connected layer as a decoder in our LSTM, TCN and

MLP models to map the extracted features to the prediction window. Temporal bundling (TB) is adopted to reduce the rate of error propagation by reducing the number of model calls (Brandstetter et al., 2022).

**Table 1: Key parameters adopted in the experiments for different models.**

| LSTM | TCN | MLP | HKAE |
|---|---|---|---|
| Input_size: 1 | Input_size: 1 | Input_size: K | Embedded Dimension: K |
| Num_layer: 3 | Num_channel: [64, 256] | Hidden_size: [64, 256] | Delay Time: 1 |
| Hidden_size: 128 | Kernel_size: 3 | Output_size: L | Bottleneck: 10 |
| Output_size: L | Drop_out: 0.1 | | Hidden_size: 16*a, a = 5 |
| | Output_size: L | | |


We evaluated the prediction results via the $R^2$ and root mean square error (RMSE) metrics. As Eq. (13) shows, the predictions are conducted in a certain time window, and the window keeps sliding to the end of this lead situation. Then, the lead time increases to adapt to a new round of sliding predictions. In this study, we calculated the difference between the predicted results and the ground truth corresponding to each case of leading prediction steps.

For leading prediction steps $j = (1, ..., L)$, we generate a forecast window $i = (1, ..., N - K - L + 1)$ and compute $R_j^2$ and $RMSE_j$ as follows:

$$R_j^2 = 1 - \frac{\sum_{i=0}^{N-K-L+1} (\widetilde{x_i^j} - x_i^j)^2}{\sum_{i=0}^{N-K-L+1} (\overline{x^j} - x_i^j)^2} \tag{11}$$

$$RMSE_j = \sqrt{\frac{1}{N - K - L + 1} \sum_{i=0}^{N-K-L+1} (\widetilde{x_i^j} - x_i^j)^2} \tag{12}$$

where $j$ is the leading prediction step in the $i - th$ forecast window. $R_j^2$ and $RMSE_j$ represent $R^2$ and $RMSE$, respectively, for

$j^{th}$ leading predictions. $\tilde{x}$, $\bar{x}$, and $x$ are the prediction, mean and ground truth of $x$, respectively.

To assess the efficacy of temporal dynamics modelling, we utilized mean-period statistics that are commonly employed in the analysis of laboratory earthquake systems (Veedu et al., 2020). We pick the shear stress peaks and compute the interval of peaks $\Delta T$ as the period of shear stress variation for the predictions and ground truth. The $R^2$ metric is used to test whether the period of stress variation of the predictions fits well. The slip interval $\Delta T$ is computed via the algorithm from Gualandi et al.

(2023), according to the timing of the shear stress peaks. The peaks are found via the *find_peak* function from *scipy*.

$$\Delta T_i = \Delta t * (t_{i+1}^{cen} - t_i^{cen}) \tag{13}$$

$$t_i^{cen} = t_i^{top} + argmin(|x(t_i^{top}), ..., x(t_{i+1}^{bot})|) \tag{14}$$

where $t_i^{cen}$ represents the central moment of slip behaviour derived from $x(t_i^{top})$, which are the maximum peaks of the shear stress series, and where $x(t_{i+1}^{bot})$ represents the minimum peaks of the shear stress series.

Then, we compute the $R^2_{slip_j}$ (or $R^2$ of the event period) for the modelled slip intervals for different leading steps:

$$R^2_{slip_j} = 1 - \frac{\sum_{i=0}^{N_{slip}} (\widetilde{\Delta T_i^J} - \Delta T_i^j)^2}{\sum_{i=0}^{N_{slip}} (\overline{\Delta T^j} - \Delta T_i^j)^2} \tag{15}$$

where $j$ is the leading prediction step and where $i$ is the number of slips in the $j^{th}$ predictions. $\widetilde{\Delta T}$, $\overline{\Delta T}$, and $\Delta T$ are the prediction, mean and ground truth of $\Delta T$, respectively. Figure 6 illustrates the computation and statistics of the slip intervals from 3 shear experiments.


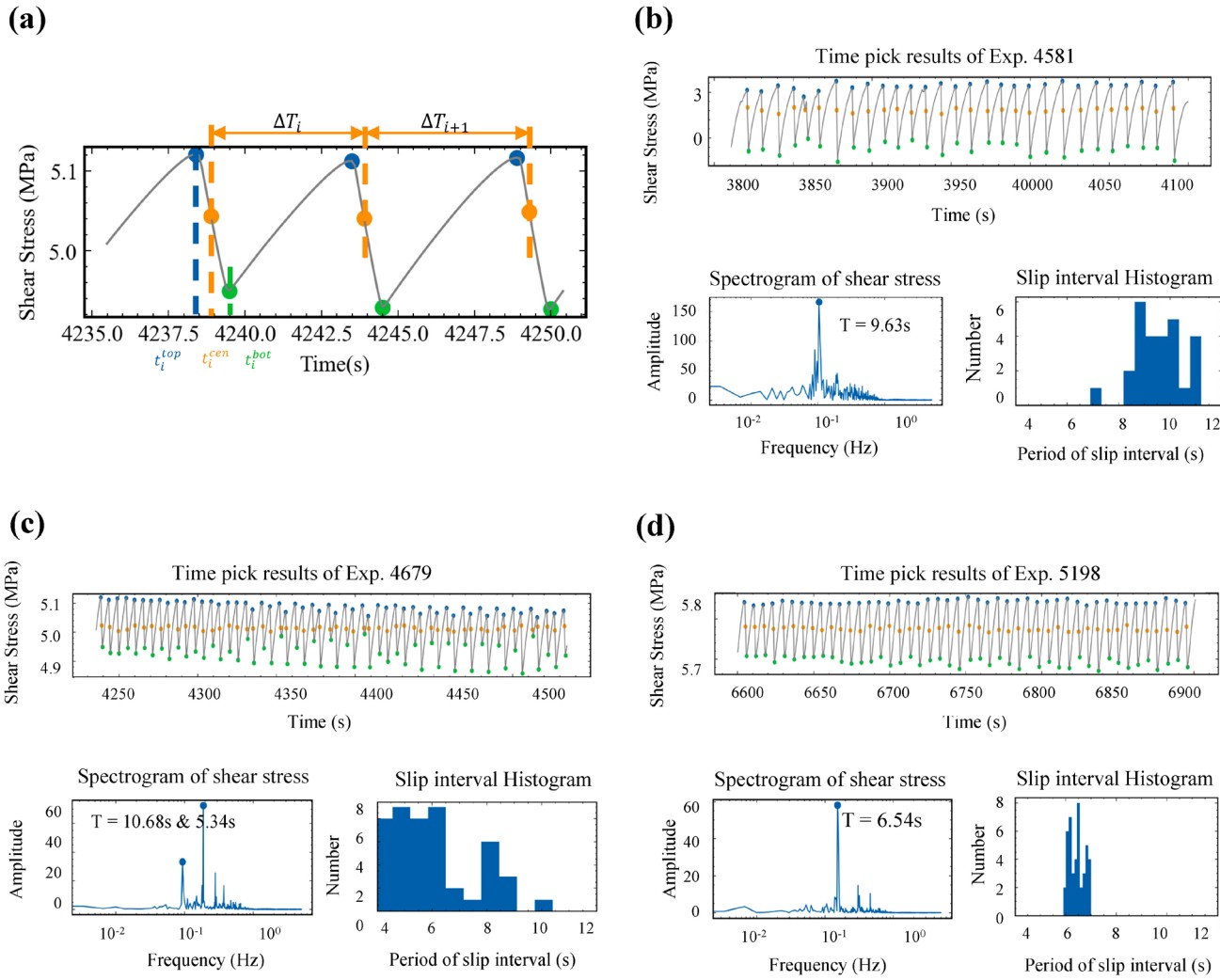

Figure 6: (a) Calculation of the slip interval **ΔT**. (b)-(d) slip intervals histogram and spectrogram of 3 experiments.

## 3 Results

### 3.1 Evaluation of the dynamic modelling process

We conduct multistep prediction experiments for three types of shear stress recorded in laboratory seismic experiments, and each type exhibits different rupture characteristics: fast rupture (Exp. 4581), alternating fast and slow rupture (Exp. 4679), and slow slip (Exp. 5198). These patterns essentially cover the types of behaviours of the laboratory slip system and correspond to real-world seismic activity and slow slip events. Unlike other statistical or deep learning methods, the HKAE is not a completely black-box approach. Thus, we can further check the status of the data during the modelling procedure to evaluate the effectiveness of HKAE modelling. We employ singular value decomposition (SVD) to illustrate the trajectory of the system and eigen-decomposition (ED) to interpret the approximated operator.

First, we adopt SVD for the result after delay embedding, utilizing three dominant right singular vector modes as coordinates to represent the reconstructed phase space (Figure 7b). The phase space describes the possible state of a dynamic system, and the reconstructed results reveal the distinct dynamic behaviours of the slip system under different slip characteristics (Stine et al., 1996). Using numerical simulation data as an example, it is evident that the system evolves along a "two-cycle-like" stable trajectory. Each cycle represents a slip process with varying intensities of stress drop, and the system alternates between these two states. The trajectories of Exps. 4581 and 5198 do not exhibit concentrated traces but rather more complex behaviours, indicative of the quasiperiodic behaviours of slow and fast slip (Gualandi et al., 2023). For Exp. 4679, the trajectory displays a more significantly disordered pattern but cycles around two main traces, illustrating the bifurcation process during the stability transition (Veedu et al., 2020). The structure of the attractors is preserved following nonlinear mapping by the HKAE (Figure 7c), indicating that the system dynamics are retained after mapping from the observed space to the Koopman subspace. We apply ED to the learned Koopman operator (Eq. (4)) and obtain the decomposed complex eigenvalue (Table 2), which represents eigendynamic modes characterized by distinct amplitudes and periods. The results are in general agreement with the dominant period obtained from frequency spectral analysis of different slip experimental stresses (Figure 6b-d). We illustrate these modes within a unit circle, which represents the attraction set. As Figure 7d shows, not all modes are near the unit circle. This may be caused by a single observation dimension or by a metastable stick–slip system in which the data do not exactly follow the attractor trajectory during the experiment (Jasperson et al., 2021). The dynamic modes represented by the eigenvalues are limited by a narrow threshold (0.01) of the distance to the unit circle. Eigenvalues that exceed the threshold are considered unstable if they are large, stable if they are small, and neutral if they are within the threshold (Avila et al., 2020). All three experiments decompose and obtain stable modes (red points, which represent modes close to the unit circle). Modes with approximately 10 s and 6 s for the cyclic fast slip occur in Exp. 4581, and those for slow slip occur in Exp. 5198. The dual modes of approximately 6 s and 10 s for the bifurcations of slips in Exp. 4679 are also modelled successfully. There are modes with amplitudes close to 1 and zero frequency in Exps. 4581 and 5198, but the amplitude of the zero-frequency (also infinite period) mode in Exp. 4679 is much lower, which means that the static component is a small percentage of the shear stress variation compared with the other 2 experiments. This may indicate that when a laboratory earthquake system is in a

state of alternating fast and slow slip, the dynamics of the system strongly vary over time, and our execution of the short-time Fourier transform (STFT) for the stress observations in Exp. 4679 should confirm this conclusion (Figure S2 in the Supplementary Information).

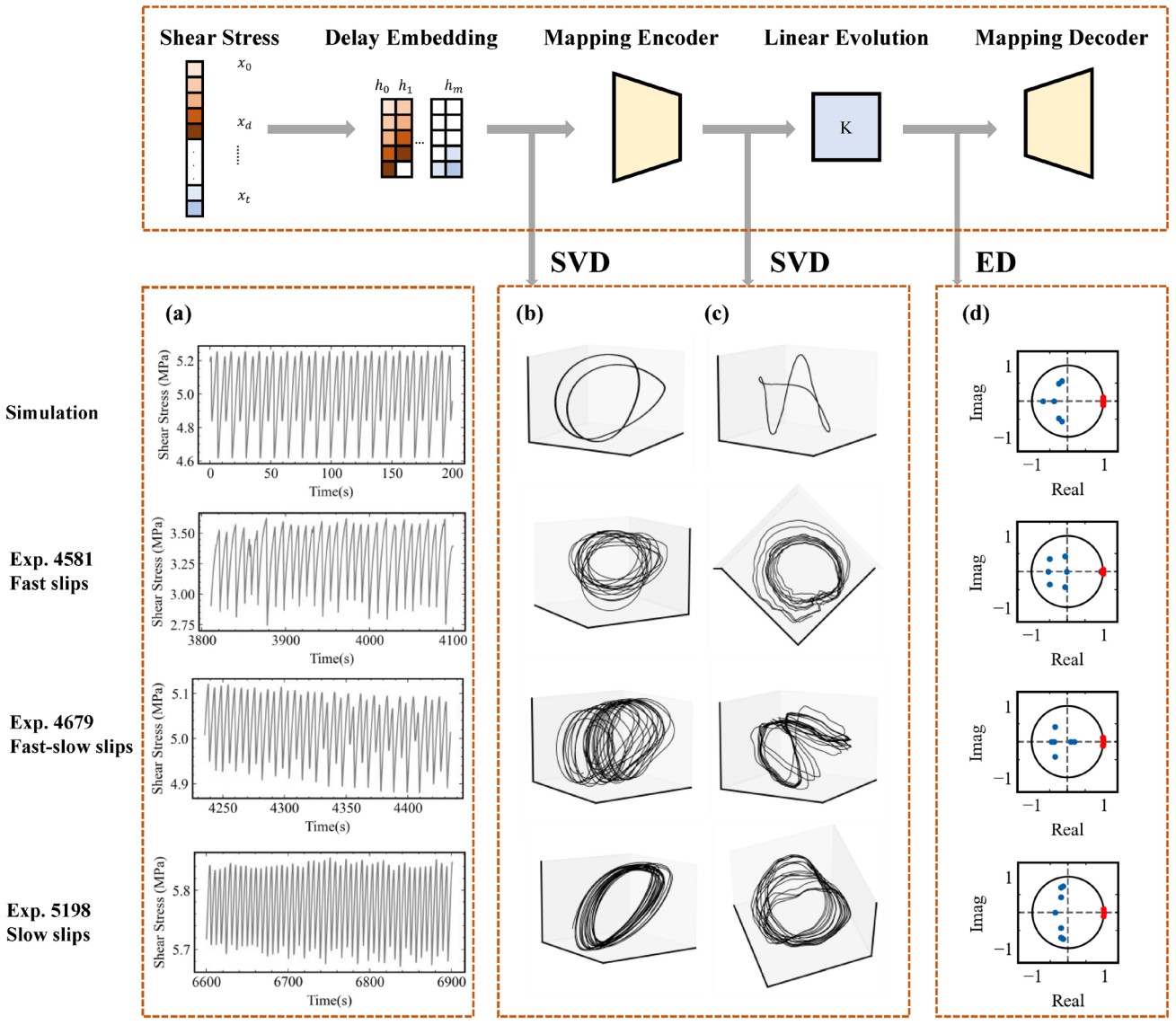

**Figure 7: Dynamic modelling steps and interpretation with the HKAE. The rows represent the different modelled experimental data. The columns represent the dynamic modelling of the data through the different steps of the model. (a) Shear stress time series from the numerical simulation and three laboratory experiments. (b)-(c) Attractor structure after delay embedding and the HKAE, using the coordinates expanded by the first three right singular vector modes. The black line illustrates the evolution of the system under three-dimensional projections. (d) Eigen decomposition of the learned Koopman operator, described by the real and imaginary parts of the complex eigenvalue. In (d), the values are directly converted to amplitude and period formats.**

**Table 2: Learned Koopman operator eigenmodes. Only the eigenvalue with positive periods is shown because of the conjugation.**

|  | Exp. 4581 | | Exp. 4679 | | Exp. 5198 | |
| # Eigenvalue | Period(s) | Amplitude | Period(s) | Amplitude | Period(s) | Amplitude |
| --- | --- | --- | --- | --- | --- | --- |
| 1 | 9.548719 | 1.001883 | 9.970778 | 0.999401 | 6.388905 | 0.994914 |
| 2 | 0.294214 | -0.071703 | 5.378387 | 0.993912 | 0.349487 | -0.13397 |
| 3 | 0.239758 | -0.500562 | 0.258224 | -0.34971 | 0.329522 | -0.20161 |
| 4 | inf | 0.967017 | inf | 0.091451 | 0.276048 | -0.195414 |
| 5 | inf | 0.94443 | inf | 0.193768 | inf | 1.00072 |

## 3.2 Evaluation of the intracycle prediction settings

We test two different experimental settings to validate the modelling and prediction capabilities pertaining to slip behaviour. The first focuses on the stress variation. Employing the historical 100 steps (spanning 10 s, which encompasses a complete slip cycle of stress increase and release), we predict the subsequent 30 steps of shear stress (spanning 3 s, which is long enough to include a process of stress increase or decrease).

Figure 8 shows that the evaluation metrics vary with the leading prediction steps for the experimental data. Generally, the classical autoregressive moving average (ARIMA) method yields competitive results during the initial 1~5 leads but severely degrades the accuracy. The next lowest performer is the TCN, which has temporal convolutional capability, followed by the MLP, the LSTM with a relatively simple structure, and the HKAE. This result is also in line with the current knowledge in the field of time series prediction, i.e., complex models are not necessarily more suitable for time series prediction tasks (Zeng et al., 2023).

For Exp. 4679, with alternating fast and slow slips, and Exp. 4581, with predominantly fast ruptures, the traditional deep learning methods appear to have poorer statistical metrics than the HKAE does after a certain number of steps ahead. The $R^2$ of the HKAE maintains a steady trend, which benefits from the dynamic modelling ability of the HKAE. According to the results of slip interval modelling, in the two experiments with more slip characteristics, the HKAE also shows robust results with increasing leading predictions. The slow slip represented by Exp. 5198, with its relatively gentle stress changes and more significant cyclic characteristics, is not too difficult from the perspective of time series prediction, so methods other than ARIMA obtain good metrics. In addition, the TCN and LSTM methods outperform the HKAE in terms of scores, and after analysing the results, we infer that this difference is related to the complexity of the slow slip system. With an embedding time of 1, we set the input length uniformly to 10 s, leading to a large embedding dimension of the HKAE, but empirically, the system dimension of the slow slip system is low (Gualandi et al., 2020; 2023). When we use a low embedding dimension, the modelling effect of the HKAE significantly improves (as shown in Figure S3 in the Supplementary Information).

In addition to evaluating the metrics for multistep lead prediction, we validate the predicted shear stress accuracy in the range of predicted 3 s. Figure 9 shows the final and future lead predictions for the test set on the basis of the three experiments. We compare the final lead predictions to demonstrate the prediction stability. For each experiment, we select two sections where

the shear stress indicates typical increase or decrease behaviour during slip. We choose LSTM, which performs consistently in different experiments, for comparison with the HKAE method. In terms of the individual lead prediction examples (right panels in Figure 9), the HKAE and LSTM have similar prediction scores, but the HKAE outputs more accurate predictions

during the stress release phase. In addition, the HKAE models the rate of stress change better. These findings suggest that the HKAE models the dynamics of laboratory rupture activity better. We validate several prediction examples for more difficult-to-predict fast slip experiments and find that the HKAE indeed models the stress release phase better (Figure S4 of the Supplementary Information). Since the stress release phase accounts for a relatively small portion of the stress change, this feature of the HKAE is masked when assessing the model performance through a sliding time window evaluation.

With respect to the timing of slip beginning, the predictions in Exp. 4581 are less accurate; there is a notable mismatch in the slip cycle and an "early release" of stresses (Figure 9a, b). This discrepancy may be attributed to the absence of creep information (Laurenti et al., 2022), as the rapid stress rupture in Exp. 4581 occurred immediately at rupture onset, in contrast to the other two experiments, where accelerated stress attenuation preceded a rapid stress drop. The HKAE estimates the stress change rate in Exp. 4581 more accurately, whereas LSTM tends to predict a lower stress release rate (corresponding to a longer

stress release time). Despite successfully modelling the rate of stress release, the HKAE tends to prematurely estimate the rupture onset. For Exp. 4679, both the HKAE and LSTM stress rates are more accurate than those for Exp. 4581. The HKAE is more accurate in estimating stress release and recovery timing, but its numerical prediction results in the initial steps are weaker than those of LSTM. Therefore, the evaluation scores are relatively low in the first few lead scores (0–0.5 s leads in Figure 8b). In addition, although the overall trend is accurately predicted using the HKAE, the numerical results are generally

high. We infer that this is a manifestation of universal operator driven modelling. We did not detrend the data, and Exp. 4679 has stronger time-varying dynamic components than the other two experiments do (Figure S2 in the Supplementary Information). The HKAE uses a global operator estimation strategy, and the presence of time-variant dynamics or multiple invariant subspaces in the theoretical linear evolving space could challenge the single universal operator's capacity to depict evolution accurately (Lan and Mezić, 2013). For Exp. 5198, the HKAE and LSTM yield similar predictions.

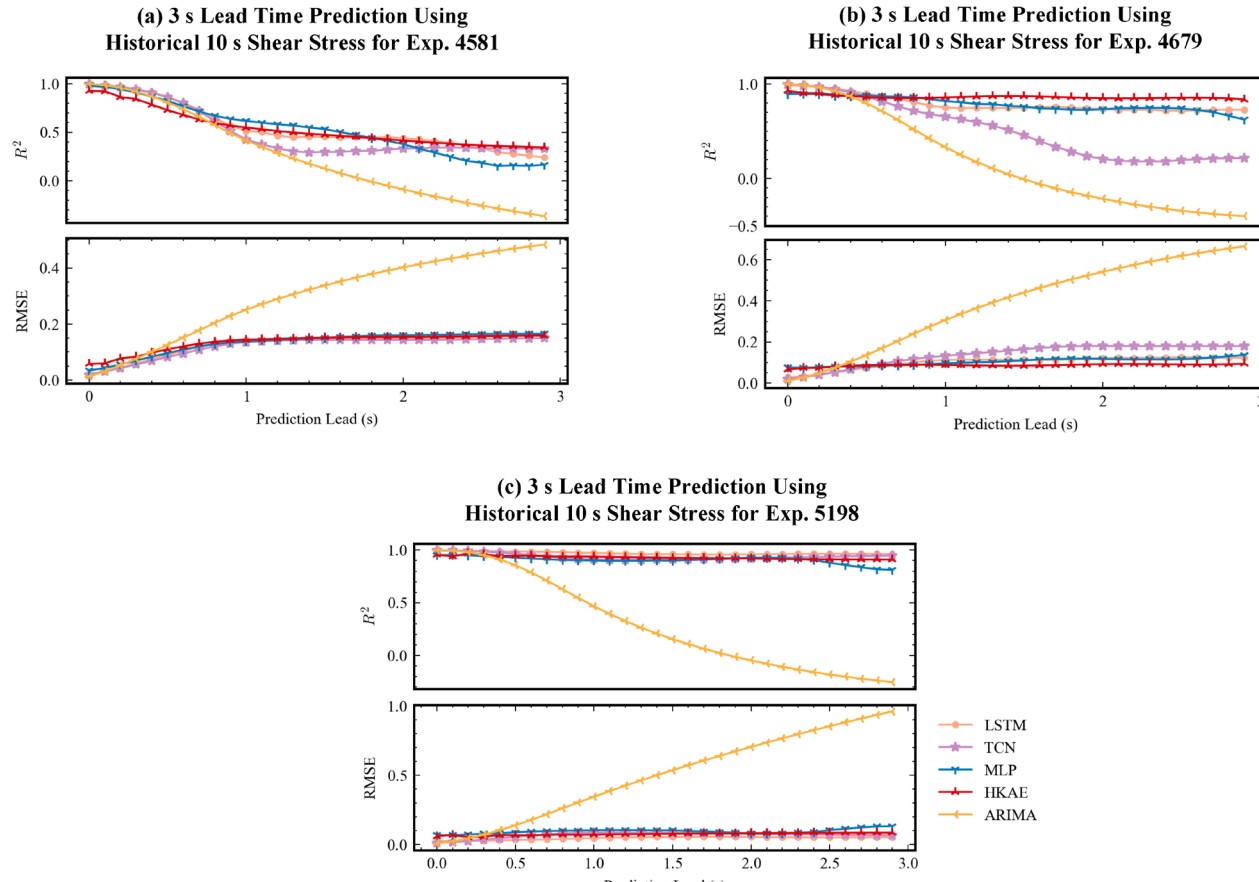

**Figure 8:** Genenral evaluation for 3 s lead prediction using historical 10 s shear stress, with R$^2$ and RMSE used as evaluation metrics. (a)-(c) for Exp. 4581, Exp. 4679 and Exp. 5198 respectively.

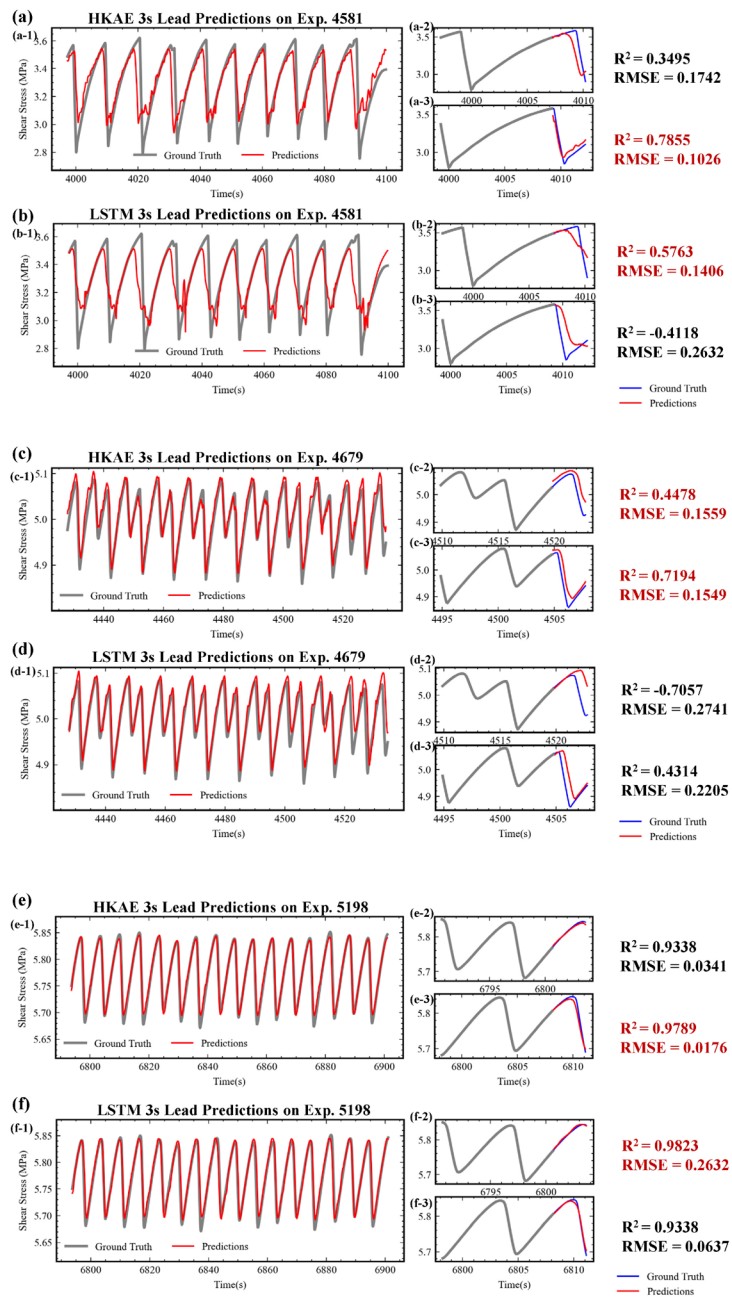

**Figure 9: 3 s leading prediction details during different phases of stress variation for different laboratory datasets.** (a), (c), and (e) show the HKAE results, whereas (b), (d), and (f) show the LSTM results. The left panels (x-1) present the total 3 s leading predictions for the test set. The right panels (x-2, x-3) illustrate the predictions in the 3 s horizon, with R² and RMSE used as evaluation metrics. Predictions with higher metrics are highlighted in red.

### 3.3 Evaluation of the intercycle prediction settings

We then extend the input to 20 s and the prediction horizon to 10 s to test the ability of the model to model the seismic cycle. This setting is the same as that used by Laurenti (2022) to test the modelling ability of slip events. Furthermore, to assess the modelling of the event cycle more intuitively, we introduce a new evaluation metric in this experiment, namely, to assess the predictive effect of the predicted results with respect to the moment of occurrence of the event.

Figure 10a-1 and 10b-1 illustrate the metrics variation with the prediction leads arising. Compared with the "10-3 s"
experimental settings (Figure 8b), when the results are predicted for the next 10 s, the slip interval scores of the HKAE predictions are higher than those of the other methods (Figure 10 a-1, b-1), which reflects HKAE has a more robust ability to capture slip system dynamics. Unlike the decrease in accuracy with an increasing number of leading steps that are evident in stress value predictions, the metrics for slip intervals do not demonstrate a steady decline. This could be attributed to the periodicity of the slip activities. Moreover, the capacity of the HKAE model to model slip behaviour more robustly indicates
its ability to capture long-term trends in the data. In terms of $R^2$ scores, the HKAE continuously models fast slip systems, especially for leading predictions after 4 s. The scores for the evaluation of the event cycle similarly support this conclusion. Fast slip systems tend to have more complex dynamic system characteristics and higher instantaneous dimensions (Gualandi et al., 2023), which demonstrates the HKAE's ability to model complex dynamic system dynamics. For Exp. 4679, the HKAE does not consistently continue its dominance when it is 1–3 s ahead of the forecast in terms of $R^2$ and RMSE scores. However,
focusing on the event cycle scores, which are more meaningful under long-term forecasts, the HKAE shows a steady advantage (Figure 10b-1).

To further assess the ability of the prediction method to model the event cycle, we counted the predictions for the slip intervals in the prediction window in the sliding prediction experiment, and assessed the goodness of fit between the predictions and the true intervals (Figure 10a-2, b-2). Considering the event cycle predictions over the entire prediction window, the HKAE
also has a advantage over the LSTM, as demonstrated by the fact that its cycle predictions are closer to the identity line. However, the $R^2$ scores of slip intervals are negative in Exp.4581, which indicates that both HKAE and LSTM have limited ability to capture the evolutionary features of fast slip systems.

For the slow slip represented by Exp. 5198, we find that the HKAE has a more stable performance in event cycle modelling (Figure S5 in the Supplementary Information). Owing to the relatively weak performance of the starting leading window,
similar to the findings of the 10–3 s experiment described above, we attribute this finding to the relatively low system dimensionality of the slow slip system (Gualandi et al., 2020; 2023).

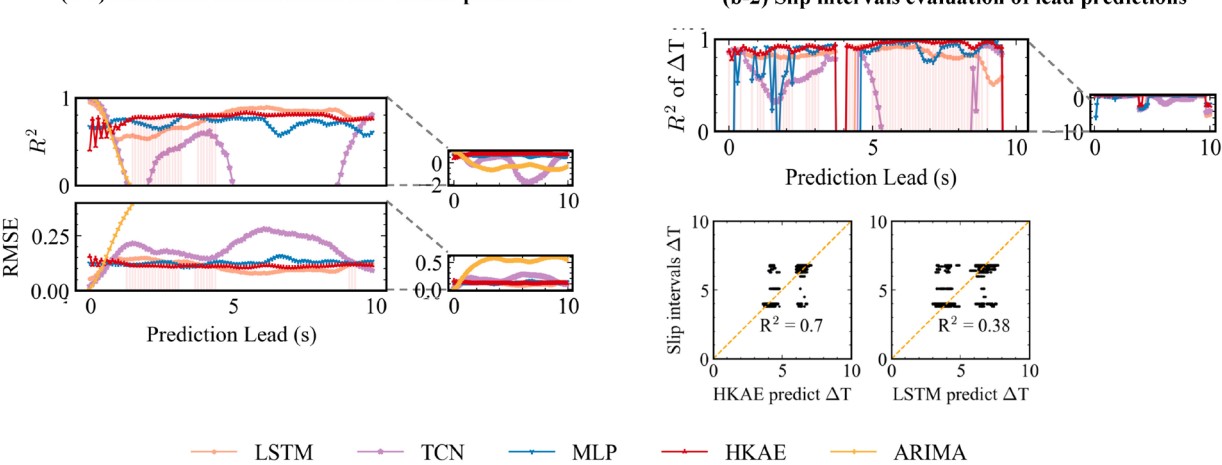

**(a) 10 s Lead Time Prediction Using Historical 20 s Shear Stress for Exp. 4581**

**(a-1) Statistical evaluation metrics of lead predictions**

**(a-2) Slip intervals evaluation of lead predictions**

**(b) 10 s Lead Time Prediction Using Historical 20 s Shear Stress for Exp. 4679**

**(b-1) Statistical evaluation metrics of lead predictions**

**(b-2) Slip intervals evaluation of lead predictions**

LSTM    TCN    MLP    HKAE    ARIMA

**Figure 10: General evaluation for 10 s lead prediction using historical 20 s shear stress, with $R^2$, RMSE and $R^2$ of event intervals (Eq. 18) used as evaluation metrics. (a)-(b) for Exp. 4581, Exp. 4679 respectively. (a-1) and (b-1) illustrate the statistical metrics ($R^2$ and RMSE) variation with prediction leads. (a-2) and (b-2) show the prediction results of the sliding prediction process for the sliding intervals were counted and compared with the real sliding intervals (Figure 6b-d).**


## 4 Discussion

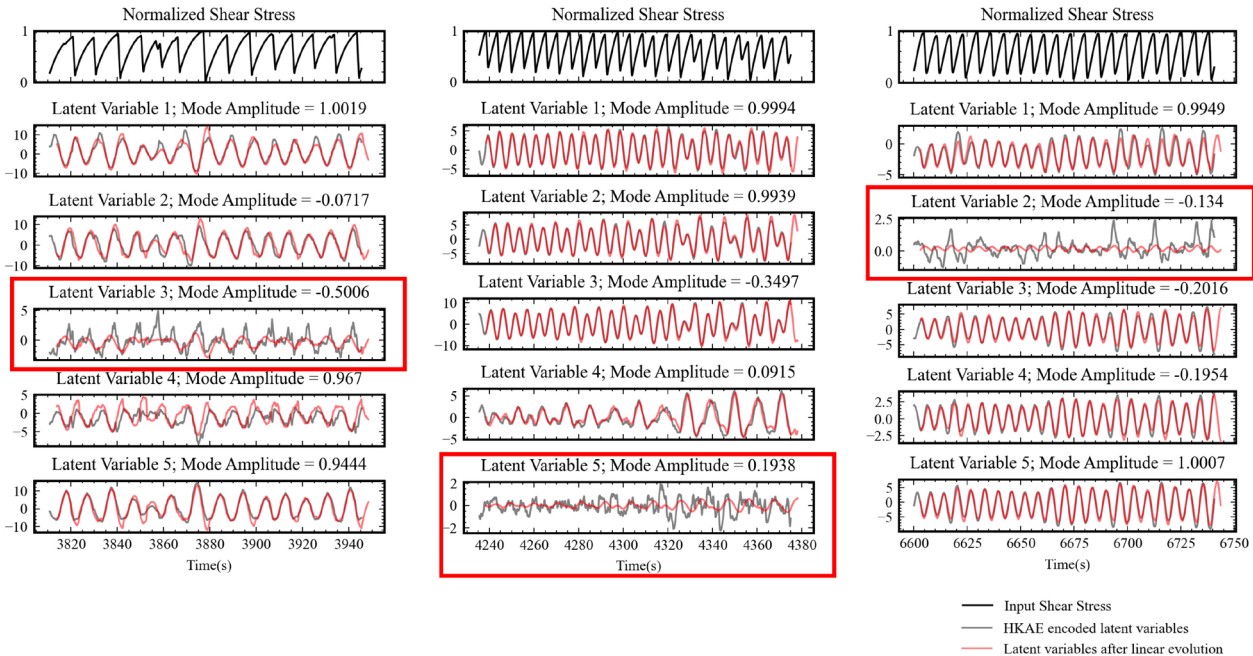

**Figure 11: Visualization of linear evolution in the latent space for 3 experiments with different slip behaviours. Input shear stress series (black), HAKE-encoded latent variables (grey) and latent variables after linear evolution (red). The red box represents the components in the latent space that are still difficult to evolve linearly.**

Modelling and predicting the behaviour of fault slip is crucial for understanding natural earthquakes. The study of natural earthquakes still presents various challenges, such as indirect observations, insufficient sampling histories of shorter fault activity cycles, etc. (Herrera et al., 2022), whereas laboratory settings provide new insights in an easier, more controllable and observable manner. Machine learning methods have been effective for accurately predicting instantaneous slip behaviour on the basis of near-term acoustic emissions (Rouet-Leduc et al., 2017; Shokouhi et al., 2021; Borate et al., 2023). However, attempts to forecast future behaviour have encountered temporal limits due to the high nonlinearity of the fault slip system in the laboratory.

To address these questions, informed by dynamic system theories, we pioneered a dynamic informed method, the HKAE, to predict the future shear stress of laboratory fault slips. The HKAE model is designed on the basis of delay embedding theory and Koopman theory and leverages the nonlinear fitting capabilities of neural networks and the systematic perspective of dynamic theories. The advantages of the HKAE include (1) multiscale modelling of laboratory slip systems under limited observations and (2) evolution mode and insights into laboratory slip from a dynamic systems perspective. The rationality of

the HKAE architecture design was further verified in the ablation experiments of the three modules, especially the setup of the Koopman Operator module (Figure S6 in Supplementary Information).

Here, we further discuss the dynamic modelling skills of the HKAE by investigating the latent space. The HKAE captures lower-complexity, more predictable dynamic components from stress observations of complex laboratory slip systems through Koopman theory, which can be visualized directly via the encoded results (Figure 11).

The scalar shear stress is reconstructed in high-dimensional phase space and then encoded by the HKAE for latent variables with dimensions equal to the approximated Koopman operator. Figure 11 illustrates different components. Considering the conjugate nature of the Koopman eigenvalues, we plot only the latent variables corresponding to half of the eigenvalues shown in Table 2. The latent variables corresponding to stable feature modes are more predictable. This is also evident from the results of the linear evolution, where the latent variables corresponding to stable modes (red line) match closely with the numerical values of the latent variables (gray line) after evolution. Moreover, there are always components such as residuals (red box in Figure 11), which, from the point of view of the HKAE, are components that are still difficult to characterize with linear dynamics after further nonlinear mapping. There could be many reasons for these hard-to-predict components, and we hypothesize that the most important reason is the lack of observations, despite the phase space reconstruction using delayed embedding. The meta-stable characteristics of laboratory slip systems also contribute (Jasperson et al., 2021). In addition, we find a greater percentage of hard-to-predict components in Exp. 4679, which is related to the unstable state of Exp. 4679. Additionally, the "residual-like" latent variable prediction for Exp. 4679 is more challenging. Considering the absence of stable, long-period modes in Exp. 4679 (Table 2), we suggest that, from the perspective of HKAE modelling, the alternating fast and slow slippage represented by Exp. 4679 represents a less stable state in the laboratory slip system than that of pure fast or slow slip. This conclusion is consistent with the understanding generated from laboratory experiments and numerical simulations, which state that rapid and slow slip alternate at the critical point of the stability transition (Veedu et al., 2020). It can also be noticed that the fast slip system (Exp. 4581) has the largest "residual-like" latent variable amplitude among the three, which is ultimately reflected in its difficulty in capturing its slip cycle variations by different prediction methods when performing long term prediction (Figure 10a). This suggests that long-term estimation of status of the fast-slip systems remains a challenging task in the presence of limited data. In addition, we find that the dimension of delayed embedding affects the prediction performance and prediction preference of HKAE to a larger extent. For example, in the slow-slip, low system dimension scenario, the overall prediction of HKAE with low embedding dimension is significantly better than that with high embedding dimension. For the fast-slow alternating slip and fast slip scenarios, the high-embedding dimension tends to obtain long-term stable prediction results, while the low-embedding dimension is able to obtain more accurate short-term prediction results (Figure 3 in the Supplementary Information). This suggests that the embedded dimension of HKAE need to be adjusted according to the slip activity state in practical applications.

The mechanisms underlying the occurrence of earthquakes caused by fast slip events, as well as slow slip events caused by slow slip, under natural conditions are still unclear. Recently, it was reported that slow slip events (SSEs) may exhibit a certain degree of numerical predictability (Gualandi et al., 2020; Truttmann et al., 2024). We suggest that the HKAE can achieve

competitive modelling of seismic activity and diagnose the dynamic behaviour of regional seismic systems by incorporating dynamic system theory. Currently there are two main challenges in the application of HKAE to actual tectonic conditions. One is that the stress state changes of actual tectonic earthquakes may be complex. In order to verify the modelling capability of HKAE, we tested it using a typical double-shear experiment representing slip fast and slow with alternation. It is shown that the stress changes are more complex under rough fault viscous slip experimental conditions (Dresen et al., 2020). Although recent studies have shown that slip Time-To-Failures (TTFs) under high roughness can be predicted using machine learning (Karimpouli et al., 2023), slip dynamical system properties under high roughness remain currently undiscussed, which may affect the future predictive performance of HKAE under such more complex conditions of slip. In addition, stress is not directly accessible under tectonic seismic environmental conditions, which increases the difficulty of applying HKAE under real conditions. However, recent studies have shown that time-series observations, such as GNSS and seismometers, exhibit the feasibility of serving as a proxy for the state change of stress in tectonic earthquakes, and this relationship is similar to that between acoustic emission signals and stress changes in laboratory earthquakes (Johnson et al., 2024; 2025). HKAE has advantages for data fusion due to its flexible neural network architecture implementation. Therefore, the generalizability of the model can be improved by integrating external data such as historical acoustic emissions or other measurable laboratory observations by means such as adding coding branches. From an algorithmic perspective, implementing the Hankel alternative view of the Koopman (HAVOK) framework (Brunton et al., 2017) and local dynamic modelling strategy (Liu et al., 2023) might further enhance the predictive capability of the HKAE.

## 5 Conclusion

Drawing upon delay embedding and Koopman theories, we propose a dynamic informed machine learning method, namely, the Hankel–Koopman autoencoder (HKAE), to achieve future predictions in complex laboratory earthquake systems. This model demonstrates competitive performance in shear stress variation and slip event modelling compared with other prediction models. To our knowledge, this is the first instance of predicting laboratory slip behaviours from a dynamic systems perspective. In addition to the modelling performance, the analysis of the execution process of the HKAE can provide dynamic diagnostics for the laboratory slip system operating behind the shear stress observations, such as those of system trajectory behaviour, characteristic dynamic modes, and system stability. HKAE prediction results and dynamical system analyses show that slip behaviour, especially the long-term future prediction of fast-slip stress states, remains challenging. Furthermore, the HKAE highlights the potential for simplifying and decomposing complex geophysical systems with a data-driven approach but combining a dynamic systems perspective. It also provides a new approach for modelling, predicting and understanding complex geophysical systems, especially those with limited observations, showing great potential in seismically monitoring and modelling the physical mechanisms of faults.

**Author contributions**

**EY:** Conceptualization, methodology, writing – original draft, writing – review & editing, formal analysis. **MQ:** Writing – review & editing, conceptualization, supervision. **LH:** Methodology, writing – review & editing. **BR**: Methodology, writing – review & editing. **SW:** Supervision, writing – review & editing. **ZD:** Supervision, writing – review & editing.

**Acknowledgements**

This research was supported by the National Key Research and Development Program of China (grant 2021YFB3900900), the National Natural Science Foundation of China (42306213) and the Deep-time Digital Earth (DDE) Big Science Program.

**Competing interests**

The authors declare that they have no conflicts of interest.

**Code and data availability**

The laboratory shear stress data are obtained from https://github.com/lauralaurenti/DNN-earthquake-prediction-forecasting and http://psudata.s3-website.us-east-2.amazonaws.com/(last access on 2024-03-21). The ODE simulation data and code are

obtained from the Open Science Framework (OSF) (at DOI 10.17605/OSF. IO/9DQH7) and https://github.com/Geolandi/labquakesde (last access at 2024--03--21). The code used to run and analyse the results of the experiments is available online (https://zenodo.org/records/13123381).

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
