# Peer review of "A dynamic informed deep learning method for future estimation of laboratory stick-slip"

_Geoscientific Model Development, 2024_

## Referee Comment (RC1)

Review for "A dynamic informed deep learning method for future estimation of laboratory stick-slip" by Yue et al.

Line 11: As far as I know, there is no actual successful application to real-world slow slip events.

Line 22: Natural fault activity is not quasi-periodic.

Line 30: Remove "firstly".

Lines 32-33: Lubbers et al. (2018) worked on laboratory earthquake: this needs to be specified because it is otherwise misleading.

Line 41: "discovered" → "showed".

Line 66: Remove "shear".

Lines 67-68: "and the equipment is used to record the changes in system properties such as pressure recorded during the shear process" is too vague. What observables are you using?

Line 69: "modified": How did you modify the model?

Line 72: To claim that the system is quasi-periodic you should run a frequency analysis. The system is likely not quasi-periodic in a strict sense, and it's more appropriate to say that it is cyclic.

Line 74: "robust" → "clear"

Figure 1: caption: remove "fault slip".

Section 2.2.1 needs references. You are largely taking the description from existing literature, and it needs to be cited.

Line 82: "fits" seems the inappropriate verb here. Maybe "described by"?

Line 83: Eq. 1 is wrong. The shear stress at time $t + 1$ is not a function of solely the shear stress at time $t$.

Line 86: The mapping $g$ is not the mapping of the Koopman operator. The wording is unclear. From eq. 2, the Koopman operator maps one step ahead the system described by the function $g$ applied to the variables that describe the system.

Lines 91-92: Technically, you need infinite modes to properly describe with a linear operator a generic non-linear dynamic.

Line 94: "are the eigenvector" and "are the eigenvalues".

Line 96: Is $j$ the time step?

Line 97: "of the k-th dynamic mode".

Line 98: "we're aware of" → "we know".

Line 98: "the mapping function".

Line 99: "the linear operator".

Line 100: You are now talking about control systems, but you do not have a controlled system in eqs. 1-3. It seems like you copy-pasted from existing literature, but it is not an accurate description of what you are actually doing.

Line 106: What are "majority observations"?

Line 118: What "As shown in Figure 3"? The sentence is incomplete. Figure 3 does not show $g$ and $K$. It is unclear what is the link with the previous sentence.

Line 121: "Lorentz" → "Lorenz". Furthermore, it needs a citation.

Eqs. 8-9: Is the running index $i$ going to $d$ instead of $n$? What is $n$? Furthermore, I think there is an error about the $j$ running index. Shouldn't it go to $L$ instead of $h$?

Line 159: "to address" → "that addresses".

Line 161: "earth" → "Earth".

Eqs. 10-12: Either you describe what $\sigma, \omega$ and $b$ are, or you do not show the equations and just refer to the literature.

Lines 192-194: You should show the results (for example, with a figure) to confirm that an embedding dimension of 100 is suggested by the method that you mention (Cao, 1997). The optimal time delay embedding is likely not 1, but you are using a value of 1 in order to build the Hankel matrix. The fact that the dimension of the Koopman operator is set to 10 further suggests that the dimensionality of the system is likely not of the order of 100, so it is difficult to understand how Cao's method could provide such a high minimal embedding dimension. The reason is likely because there is a lot of redundant information with the small time delay embedding of 1.

Line 197: How are they adapted? Just by trial and error? Or with some consistent procedure?

Lines 197-199: What is a multi-step trick?

Tab. 1: adopted instead of adapted? These parameters are fixed, not adapting.

Eq. 14: Is the mean the mean over the whole time series or in the window used to make the forecast? Using the whole time series mean seems wrong to me: why should the forecast tend to the mean of the whole time series when using a specific window that has a subset of the whole time series?

Eq. 17: The notation is quite odd. What you name "slip" or "stress" it's actually just the index, so it is a time. When writing "slip_center_index" what is typically understood is that that is the value taken by the slip at the time index given by the index. Furthermore, "the central moment of a slip behaviour" is not clear. What is a "slip behaviour"?

Eq. 18: The fraction and parenthesis are likely wrong: check the consistency of your equation.

Lines 237-239: SVD is not applied to validate the effectiveness of the dynamic modelling.

Lines 243-244: Is it really quasi-periodic?

Lines 249-253: The wording is very cryptic. What do you mean by "a single observation dimension"? Furthermore, a dynamical system will follow a trajectory that defines the attractor. How is it possible to have an unstable stick-slip system where the data does not exactly follow the attractor trajectory during the experiment?

Line 257: Experiment p5198 does not have a stable mode at 9 s.

Lines 259-260: Why a zero-frequency mode with low amplitude is representative of a strong time-varying component? What does it mean "strong time-varying component"?

Line 278: Why only two experiment settings?

Line 284: You claim that the HKAE results are clearly better when using the DeltaT evaluation metric. OK, it is better, but the R2 is most of the time negative, and, in the best situation, slightly positive. This seems to be a very poor result for all the architectures. Even if for the HKAE it is better, it is still a poor result.

Line 285-287: The sentence is unclear. It is surprising that the RMSE and R2 metrics are worse for the HKAE in the first few forecasted steps. This would suggest that for short term predictions LSTM and TCN are better. But the dynamics that you are extracting should degrade with longer lead prediction times. Something that instead is not really observed.

Lines 289-294: Which metric? You tested more than one. The setting 20-10 s is shown in Fig. 8b, not 8a. Finally, how can you claim that the lead prediction can cover 1-2 complete cycles? When we look at Fig. 8b, the DeltaT metric shows that many points are not on the diagonal, meaning that the prediction is wrong in many cases regarding the timing of the events. This class misrepresentation (DeltaT small or large are the two classes) indicates that your lead prediction cannot cover multiple cycles.

Line 296: If the slip events were periodic, it would be very easy to predict them, and you should have zero error. The events are not periodic.

Lines 318-322: There is confusion on what you mean by underestimation. For example, if you say that the minimum value is underestimated, it means that the predicted shear stress is smaller than the actual minimum shear stress. But the figure shows that the predicted minimum shear stress is higher. What you mean is that the stress drop is underestimated. Please be consistent and describe properly the results.

Line 326: "moment of " → "instant for".

Line 343-346: I disagree with the way you are reading the figure. In Exp. 4679 LSTM clearly has a better RMSE for most of the lead time between 3 and 10 s (non-grey area). It is not even clear how you are drawing those lines. If you look at just the points (and not the interpolated line, which is not an actual result), LSTM is in fact comparable to HKAE until 4.5 s, and then always better or with same results. For Exp. 4581 the RMSE is basically the same for all algorithms. Finally, for Exp. 5198 the RMSE of LSTM is the best until about 7 s, where it becomes like the one of HKAE, and only slightly worse after 9 s. But what does it mean to be able to have a slightly better forecast at 9-10 s lead time? Based on your results, if I had to pick an algorithm to make earthquake forecasts, I would use LSTM: it clearly outperforms HKAE in the forecast of the immediate next steps, which is what matters the most. For long-term forecasts (like 9-10 s, which is a full cycle ahead), one can very likely use purely statistical methods and obtain similar results.

Lines 369-370: The two "stable" dynamics clearly show an amplification over time, which makes them not as stable as you depict them. From Fig. 11b it is not clear what they represent. What do you mean with "input"? These are modes obtained after processing the input data in some way.

Lines 387-388: This is not true. For short-term predictions HKAE seems to perform poorly compared to other algorithms. If we look at either Fig. 8 or Fig. 10, we see that for the first few time-steps of forecast the HKAE is consistently not the best, with LSTM being often the best, followed by TCN.

Lines 392-394: This is not true. On many occasions HKAE was not showing superior performance in terms of prediction accuracy.

Figure 2: The caption is too synthetic. You need to explain what the panels represent. You need to add letters to the various panels. All the panels need labels for the axes. You are using the Lorenz63 system time series, but never mentioned it in the text. If you are using the figure taken from another publication, you need to state it and you need to have the permission to reproduce it here.

Figure 3: I find the usage of the Lorenz system as an example not appropriate. The Lorenz system is not the focus of this work, and you could (and, in my opinion, should) use the dynamical system that mimics the laboratory earthquakes instead.

Figure 4: My understanding is that the output of the Koopman operator applied to $g(x_t)$ is $g(x_{t+1})$, so this should be the reconstruction in the embedding space. From your figure this reconstruction comes after the application of $g^{-1}(\cdot)$, but this should give you a signal back in the original input space (and indeed you have an arrow coming back down as "One-step Evolution"). I think that this figure is inaccurate and needs to be corrected.

Figure 8: In the main text you start from experiment 5198, and this should be the first to appear in the figure (i.e., on the left). Furthermore, you have also shown ODE simulations earlier on. You should show the results on the simulations as well. They are periodic, so you should be able to recover them perfectly. If not, this would cast some doubts on the procedure.

---

## Author Response (AR1)

**Reply to Review for "A dynamic informed deep learning method for future estimation of laboratory stick- slip" by Yue et al.**

**[Review]:**

Yue et al. developed an algorithm (HKAE) to perform time series forecasting. The algorithm exploits concepts derived from dynamical systems theory and Koopman theory, and it uses an autoencoder architecture to realise the link between the two. They decided to apply the algorithm to laboratory earthquakes data, and in particular to shear stress time series.

I think that the idea and the results are interesting. Nonetheless, there are several points that need to be better explained and/or further developed.

I have two major concerns about the manuscript.

1) The first problem that I see comes from your interpretations. You state that your algorithm outperforms the existing ones (e.g., you tested LSTM, TCN and MLP). But in many occasions this is not true. Your Figures 8 and 10 show that, especially for the immediate next future, LSTM performs better than HKAE. I would recommend you to not oversell the HKAE algorithm.

2) The second problem comes from the pre-processing of the data. Reading the code, I noted that there is a pre-processing step to smooth the data. In the manuscript you do not mention any filtering or smoothing step. The smoothing function that you use is taken from the *statsmodels* package, and it takes the closest data to perform a local linear regression. The closest data in a 1-dim time series can come from both the past and the future. This means that when you smooth the data you are introducing information from the future. This is a problem if you want to evaluate forecasting performances. You need to clarify how many data from the future are used to smooth the data.

Other two, less critical but still important, problems are the following.

One concerns the reproducibility. In order to reproduce the results, it is important that you add a README.txt file in your repository. Furthermore, you should add a requirements.txt file with the details of the packages that you used. It is a good practice to do it so that people can create a local virtual environment and reproduce your results. Some comments in your code are not in English, and you should translate them.

Finally, it is not easy to follow the reasoning and the various steps mainly because of the overall poor English structure. I was trying to write down the correction myself, but after line 70 I gave up because there were too many corrections to suggest. In the Acknowledgments section you mention that the manuscript was polished with GPT-4. Sometimes the feeling is that entire paragraphs were written automatically, without a proper logical connection with the next part. I highly recommend you ask a native English speaker to review and edit the manuscript.

For detailed comments, please see the attached file.

**[Response]:**

Thank you for your affirmation of our ideas and for your very detailed review comments, which are extremely helpful for improving our results. In response to your review comments, we will make the following responses:

First about your two major concerns:

1. The most important feature of HKAE is that it considers laboratory earthquake slip from the perspective of dynamical system, rather than using purely statistical methods or black-box deep learning methods. Specifically, it achieves:

   (1) **Multi-scale modeling of laboratory slip system under limited observations:** By modeling the dynamics of the HKAE, we are able to obtain estimates of the future state of the system in two aspects: between slip cycles, the HKAE is able to better estimate the period of the events, shown in Figure 8; and get more accurate estimate of stress variation within the slip cycle, especially during the stress-release phase, shown in Figure 9. Noticed that we only use shear stress series to reconstruct the system, which will be a potential characteristic when the observations are scarce in field.

   (2) **Interpretability and insights of laboratory slip from a dynamical system perspective:** By approximating the Koopman operator, we are able to further understand the dynamics of laboratory stick-slip. From the amplitudes and periods of dynamic modes obtained from the HKAE, it can be found that there are stable dynamical modes in all the meta-stable slip systems. and the systems with pure fast slips and slow slips are more stable. In terms of specific modes, the systems with fast slip and slow slip are more stable than those with alternating fast and slow slip, which is consistent with the previous understanding obtained for laboratory slip systems under different normal stress loadings.

   Based on your comments, we realize that there is some ambiguity in our statement, and we adjusted it in the revised version.

2. Regarding the data smoothing issue, we must clarify that we did not use data smoothing in our actual experiments. Instead, we directly used the preprocessing method in Laurenti's work (2022), where we resampled the stress sampling data from 0.001s to 0.1s, taking only the first point of the 100 sampling points as the resampled result, thus avoiding information leakage. The smoothing part retained in the code is from our historical processing, and we have removed it in the updated code version.

Then in response to your other two concerns:

1. Thank you very much for emphasizing reproducibility. In the updated code version, we have added a requirements document and included a README for quick experimental reproduction. The comments in the code have also been fully translated into English. The new URL of our code is: https://zenodo.org/records/12627258.

2. Thank you very much for your detailed correction suggestions. From your attachment, we can see that you have provided very detailed comments on the entire manuscript, which greatly helps improve our results and manuscripts. Once again, thank you for your patience. Additionally, we apologize for the somewhat poor language expression. Considering GPT-4's excellent performance in language polishing, we used it to refine our manuscript, but it may have altered the original meaning during content adjustments. In the

revised version, we have comprehensively checked the content expression and had it re-polished. However, the current interactive response does not allow for the submission of a revised version, so we have provided partial responses to the textual portions of your queries within the response manuscript.

Once again, thank you for your detailed and fair review comments. For the detailed review comments in your attachment, our responses are as follows:

(Black bold text: Reviewers' comments; Purple text: Our responses; Red text: changes in manuscript)

Reference:

[1] Laurenti, L., Tinti, E., Galasso, F., Franco, L., and Marone, C.: Deep learning for laboratory earthquake prediction and autoregressive forecasting of fault zone stress, Earth and Planetary Science Letters, https://doi.org/10.1016/j.epsl.2022.117825,2022.

**Section 1: Abstract & Introduction**

**1. Line 11: As far as I know, there is no actual successful application to real-world slow slip events.**

[Response]:

Thank you for your valuable suggestion. Upon checking of recent work, we have turned our statements into "Recently, machine learning methods have been proven effective in predicting instantaneous fault stress in laboratory settings and fault activities on Earth".

(1) The point we intended to convey is that machine learning has been successfully applied to the prediction of stress changes, using GNSS signals as a proxy under natural conditions (Rouet-Leduc et al., 2019). Also, the long-term seismicity variation (Velasco et al., 2022).

(2) We have also noted recent work that employs data-driven methods such as EnKF to estimate future slow slip events (Hamed et al., 2023). However, it is true that there are currently no cases of deep learning being applied to SSEs in real-world scenarios.

Reference:

[1] Rouet-Leduc, B., Hulbert, C., and Johnson, P. A.: Continuous chatter of the Cascadia subduction zone revealed by machine learning, Nature Geoscience, 75–79, 2019.

[2] Velasco Herrera, V. M., Rossello, E. A., Orgeira, M. J., et al.: Long-term forecasting of strong earthquakes in North America, South America, Japan, southern China and northern India with machine learning, Frontiers in Earth Science, 10, 905792, 2022.

[3] Diab-Montero, H. A., Li, M., van Dinther, Y., et al.: Estimating the occurrence of slow slip events and earthquakes with an ensemble Kalman filter, Geophysical Journal International, 234(3), 1701-1721, 2023.

**2. Line 22: Natural fault activity is not quasi-periodic.**

[Response]:

Thank you for your valuable suggestion. We changed our statements into "The capability of HKAE
to decompose and model complex temporal dynamics highlights its potential in and sparse-observed geophysical system with cyclic characteristics like natural fault activities".

**3. Line 30: Remove "firstly".**

[Response]:

Thanks to your suggestion, we have removed "firstly" from the revised version.

**4. Lines 32-33: Lubbers et al. (2018) worked on laboratory earthquake: this needs to be specified because it is otherwise misleading.**

[Response]:

Thanks to your comments, we have added the following scope of work for Lubbers: "Lubbers et al. (2018) found that the event catalogue, which is more available in natural earthquake, can also predict the transient fault mechanism in laboratory earthquake."

**5. Line 41: "discovered" -> "showed".**

**[Response]:**

Thank you for your valuable suggestion. We changed our statements into "Laurenti et al. (2022) showed that laboratory fault zone stress can be inferred autoregressively."

**Section 2: Materials, methods and models**

**1.  Line 66: Remove "shear".**

**[Response]:**

Thanks for your suggestion, we have removed "shear" in the revised version.

**2.  Lines 67-68: "and the equipment is used to record the changes in system properties such as pressure recorded during the shear process" is too vague. What observables are you using?**

**[Response]:**

(1)  According to the original data records, the actual recorded mechanism data including "Normal stress", "Shear stress", "displacement" etc. (Table 1). Here, we only used the term "Shear Stress".

(2)  We add the observables during the experiments and changed our statements into "…and the equipment is used to record the mechanism changes (SI. Table 1) in system during the shear process (Figure 1a-b)."

**Table 1:** Observables in shear experiments.

| No. | Observables | Dimension (unit) |
|-----|-------------|------------------|
| 1 | LP Displacement | mic |
| 2 | Shear Stress | MPa |
| 3 | Normal Displacement | Micron |
| 4 | Normal Stress | MPa |
| 5 | Friction Coefficient (Mu) | / |
| 6 | Layer Thickness | Mircon |
| 7 | Sample Frequency | Hz |
| 8 | Time | sec |

**3.  Line 69: "modified": How did you modify the model?**

**[Response]:**

Here "modified" means that the equations used in the work of Gualandi (2023) are modified based on the original Rate-State-Function model. To avoid ambiguity, we have adapted the statement here to:

"We derive the numerical simulation data from the model (Figure 1d) in Gualandi's work (2023)".

**4.  Line 72: To claim that the system is quasi-periodic you should run a frequency analysis. The system is likely not quasi-periodic in a strict sense, and it's more appropriate to say that it is cyclic.**

**[Response]:**

Thank you for your suggestion. We agree that the system is not quasi-periodic in a strict sense or mathematically.

(1)  Firstly, the "quasi-periodic" characterization of laboratory earthquakes is discussed in the Veedu's work

(2020). Here we refer the key figure of their work as follows, which showed the quasi-periodic characteristic of slow and fast slip events.

[Figure]

**Figure 1.** part of Figure. 4 from Veedu's work (2020)

(2) To further explain the "quasi-periodic" feature of data used in our experiments, we supplemented the data with time-domain and frequency analysis. Specifically, we follow the approach in Fig. 2 to find the highest and lowest points of the stress change, and use their midpoints as the time point at which the event occurs, defining the time to next event as Period T. We analyze all three data in this way in the time domain, and obtain a plot of the distribution of event periods in the data from the three experiments (left-down figure in panel(b)-(d)). In the frequency domain we performed a Fourier analysis (right-down figure in panel(b)-(d)) on all three data and picked the peaks from the spectrum and calculated the corresponding period values. In the frequency domain, it can be found that for 3 experiments with different slip behaviours they have dominant frequencies. From the time domain, it can be found that there will be a certain period range with prominent number of event periods. Thus, we adopt "quasi-periodic" here and in later part.

[Figure]

**Figure 2.** (a) Event instant and period picking rules. (b)-(d) event period histogram and frequency analysis for 3 experiments.

Reference:

[1] Veedu, D., Giorgetti, C., Scuderi, MarcoM., Barbot, S., Marone, C., and Collettini, C.: Bifurcations at the Stability Transition of Earthquake Faulting, Geophysical Research Letters, https://doi.org/10.1029/2020GL087985, 2020.

5. **Line 74: "robust" ->"clear"**

**[Response]:**

Thank you for your valuable suggestion. We changed our statements into:

"We used the simple simulation of the data as an additional validation (Figure 1d), which shows a clear switch between two slip behaviours with stress variation."

6. **Figure 1: caption: remove "fault slip".**

**[Response]:**

Thank you for your valuable suggestion. We have removed "fault slip" of Fig. 1's caption in the revised version.

7. **Line 82: "fits" seems the inappropriate verb here. Maybe "described by"?**

   **Line 83: Eq. 1 is wrong. The shear stress at time $t+1$ is not a function of solely the shear stress at time $t$.**

**[Response]:**

Thank you for your valuable suggestion. We reclaimed the representation of state, observation, shear stress and their relationship in text and equations.

What we are trying to convey here is that the state $x$ of the laboratory seismic system can be viewed as a dynamic system change, and the shear stress can be viewed as an observation of the state. We change the statements into:

"Laboratory earthquake can be conceptualized as being governed by a dynamical system, described by Eq. (1)".

$$s_{t+1} = F(s_t) \tag{1}$$

$s_t$ is the state of laboratory slip system, $F$ presents the governing function of the system. The shear stress $x_t$ can be viewed as an observation of the laboratory slip system state $s_t$."

8. **Line 86: The mapping $g$ is not the mapping of the Koopman operator. The wording is unclear. From eq. 2, the Koopman operator maps one step ahead the system described by the function $g$ applied to the variables that describe the system.**

**[Response]:**

Thank you for your comments. We have changed the text in the revised version.

Here we want to claim that $g$ is a mapping from the observation mapping to the Koopman operator performing space where it is located, the current expression is not proper and we change it to "It states that all finite-dimensional nonlinear systems can evolve in an alternative space through the mapping $g$ and the infinite-dimensional Koopman operator $K$." in the updated version.

9. **Lines 91-92: Technically, you need infinite modes to properly describe with a linear operator a generic non-linear dynamic.**

**[Response]:**

Thank you for your comments.

Theoretically accurate Koopman operators are infinite dimensional. The current related methods around Koopman operators, such as DMD, Deep Koopman methods, found that the finite-dimensional linear matrices are able to efficiently approximate the infinite-dimensional Koopman operators in applications.

Here, we also emphasize that our operator is approximate ("the learned approximate Koopman operator" in Line 91 of manuscript). We added a note on the approximate Koopman operator in the manuscript to avoid

misunderstandings.

**10. Line 94: "are the eigenvector" and "are the eigenvalues".**

**[Response]:**

Thank you for your valuable suggestion. We change the words into "eigenvectors" and "eigenvalues" in the revised version.

**11. Line 96: Is $j$ the time step?**

  **Line 97: "of the k-th dynamic mode".**

**[Response]:**

Thank you for your valuable suggestion. Yes, $j$ here represents time step.

We have added and adjusted the notation of eq. 5 as follows:

"…

$$b_j^k = b_0^k e^{\frac{j}{\Delta t} \log \lambda_i} \tag{5}$$

Here $\boldsymbol{b^k} = [b_1^k, b_2^k, \dots, b_N^k]$ represents the temporal evolution of $k - th$ dynamic modes, and $j$ represents the time step…"

**12. Line 98: "we're aware of" -> "we know";**

  **Line 98: "the mapping function";**

  **Line 99: "the linear operator".**

**[Response]:**

Thank you for your valuable suggestion. We have changed these words in the revised version.

**13. Section 2.2.1 needs references. You are largely taking the description from existing literature, and it needs to be cited.**

  **Line 100: You are now talking about control systems, but you do not have a controlled system in eqs. 1-3. It seems like you copy-pasted from existing literature, but it is not an accurate description of what you are actually doing.**

**[Response]:**

Thank you for your valuable suggestion. We have added related references of Koopman theory and related applications in the revised version.

"Koopman theory is a mathematical theoretical framework. It states that all finite-dimensional nonlinear systems can evolve in an alternative space through the mapping g of the infinite-dimensional Koopman operator K (Koopman, 1931; Brunton et al., 2021)."

"…for example, to explore what are the main driving components in the evolution of dynamical systems (Brunton et al., 2021), what is the pattern of its growth (Schmid et al., 2010), and so on."

**14. Figure 2: The caption is too synthetic. You need to explain what the panels represent. You need to add letters to the various panels. All the panels need labels for the axes. You are using the Lorenz63 system time series, but never mentioned it in the text. If you are using the figure taken from another publication, you need to state it and you need to have the permission to reproduce it here.**

**[Response]:**

Thank you for your valuable suggestion. Here we are trying to reflect the conceptual design of Koopman's theory in section 2.2.1 to make it easier understanding.

Based on your suggestions, we have refined the Figure 2 as follows:

(1) We add the panel letters and captions respectively.

(2) We add the Lorenz system and formal reference in the caption text.

[Figure]

**Figure 3.** Replotted Fig.2 in manuscript.

Illustration of the transformation between nonlinear trajectory of high dimensional state with $F$ (a) and a linear operator $K$ represented dynamics (b), taking Lorenz63 system for example. Lorenz63 system observations are used for representation the transformation (Lorenz, 1963).

**15. Line 106: What are "majority observations"?**

**[Response]:**

Thank you for your valuable suggestion. The "majority observations" represents the "observations that carry a lot of information about system state changing".

Here we would like to express the observations that contain major information about state variations, and "majority observations" may cause misunderstanding, so in the updated version, we change the expression to:

"For analyzing dynamical systems using Koopman theory, it would be nice to have states directly of the system, or observables that carry information about the main changes in the system. However, in the real world we often obtain state quantities with a limited signal-to-noise ratio, or observations that do not carry all the information about the changes in the system, i.e., partial observations. Here, inferring the state change of a laboratory slip system from the shear stress can be viewed as inferring the future evolution of the system from a very limited number of partial observations. "

**16. Line 118: What "As shown in Figure 3"? The sentence is incomplete. Figure 3 does not show $g$ and $K$. It is unclear what is the link with the previous sentence.**

**Figure 3: I find the usage of the Lorenz system as an example not appropriate. The Lorenz system is**

**not the focus of this work, and you could (and, in my opinion, should) use the dynamical system that mimics the laboratory earthquakes instead.**

**[Response]:**

Thank you for your valuable and justified suggestions.

(1) We rewrite this sentence and adjust the notation in Figure 2、3, Eq.6 and text for consistency.

(2) We use the laboratory earthquake system synthetics instead in revised Figure 3 refer to Gualandi's work (2023). We use the Lorenz system considering its widely recognized representation of the dynamical system. It makes more sense for laboratory earthquake system appear in this manuscript.

"Figure 3 represents the process of delayed embedding."

[Figure]

**Figure 4.** Revised Fig 3 in the manuscript.

17. **Line 121: "Lorentz" -> "Lorenz". Furthermore, it needs a citation.**

**[Response]:**

Thank you for your valuable suggestion. We change the text and add the citation of Lorenz System (Lorenz, 1963).

18. **Figure 4: My understanding is that the output of the Koopman operator applied to $g(x_t)$ is $g(x_{t+1})$, so this should be the reconstruction in the embedding space. From your figure this reconstruction comes after the application of $g^{-1}(\cdot)$, but this should give you a signal back in the original input space (and indeed you have an arrow coming back down as "One-step Evolution"). I think that this figure is inaccurate and needs to be corrected.**

**[Response]:**

Thank you for your feedback. We replotted the Figure 4 for clear expression of HKAE model.

(1) Actually, the Figure 4 is organized based on the design of loss function. First of all, it's true that the output of Koopman operator module applied to $g(x_t)$ is $g(x_{t+1})$. Here the "Reconstruction" in Figure 4 represents the procedure **without applying Koopman operator**. By this, we can compute the

"Reconstruction loss", to make sure that the mapping pair $g$ and $g^{-1}$ can achieve mapping between embedding space and original space. The output of this procedure is actually the input itself, so we didn't represent it in the figure.

(2) The narrow below $g$ shows another procedure with applying Koopman operator, which actually achieve future estimation, and further get the lead predictions as outputs (as Figure 4 shows).

(3) We replotted the figure as follows:

[Figure]

**Figure 5.** Modified Figure 4 in manuscript.

**19. Lines 197-199: What is a multi-step trick?**

**Eqs. 8-9: Is the running index $i$ going to $d$ instead of $n$? What is $n$? Furthermore, I think there is an error about the $j$ running index. Shouldn't it go to $L$ instead of $h$?**

**[Response]:**

Thank you for your comments. We corrected the equations and text in the revised version.

(1) $i$ represents running index in historical input $L$ (also $n$ in Eq. 7, and embedded dimension $d$ when delay time $\tau = 1$).

(2) $j$ represents prediction steps in forecast horizon $K$. The notation $h$ is not correct in the Eq.8.

(3) The multi-step trick refers to the fact that for each sliding window, we make $L$-step future predictions for all time index within the entire window, instead of only predicting $L$-step in the future for the last moment within the sliding window. Thus, each sliding window in our loss term computes all steps ($L$) of the multi-step prediction ($K$) results, which is why Eq. 8 is summed for twice.

(4) For consistency, we adjusted the Eqs.7-8 and Figure 4 in the revised version:

$$\varepsilon_{reconstruct} = \frac{1}{2L} \sum_{i=1}^{L} \| h_i - \widehat{h}_\iota \|_2^2 \tag{7}$$

$$\varepsilon_{evolution} = \frac{1}{2KL} \sum_{j=1}^{K} \sum_{i=1}^{L} \| h_{i+j} - \widetilde{h}_{i+j} \|_2^2 \tag{8}$$

**20. Line 159: "to address" -> "that addresses".**

**Line 161: "earth" -> "Earth".**

**[Response]:**

Thank you for your valuable suggestion. We changed these words in the revised version.

**21. Eqs. 10-12: Either you describe what $\sigma$, $\omega$ and $b$ are, or you do not show the equations and just refer to the literature.**

**[Response]:**

Thank you for your valuable suggestion. Considering the generality of LSTM in time series modeling, we removed Eqs. 10-12 in the revised version.

**22. Lines 192-194: You should show the results (for example, with a figure) to confirm that an embedding dimension of 100 is suggested by the method that you mention (Cao, 1997). The optimal time delay embedding is likely not 1, but you are using a value of 1 in order to build the Hankel matrix. The fact that the dimension of the Koopman operator is set to 10 further suggests that the dimensionality of the system is likely not of the order of 100, so it is difficult to understand how Cao's method could provide such a high minimal embedding dimension. The reason is likely because there is a lot of redundant information with the small time delay embedding of 1.**

**[Response]:**

Thank you for your valuable suggestion.

(1) The parameters we ultimately adopt are not confirmed by Cao's method, and we refined the statement of parameter selection to:

"For the delay time, we took 1, on the one hand to construct the Hankel matrix to satisfy the prediction requirements, and on the other hand Brunton et al. (2017) reported that based on practice, they found that most of the systems have a delay time of 1. With a delay time setting of 1, the embedding dimensions in our model architecture are numerically equal to the number of steps in which the historical information is used. Given the experimental setup in Laurenti's work (2022), we suggest that sufficient historical information is needed to predict future changes. To evaluate both intra- and inter-cycle scenarios of slip, we set up an experimental setup using 20s of historical data to predict 10s in the future and 10s of historical data to predict 3s in the future, corresponding to embedded dimensions with 200 and 100."

(2) We tested Cao's method to generate theoretically optimized parameters as Table 2 shows. It's true that Cao's method gives a recommendation embedded dimension at 11 when delay time $\tau = 1$, which is similar to the dimension of Koopman operator we adopted in HKAE.

As Figure 3 illustrated, we fixed the delay time $\tau = 1$, and test 30 step lead forecast with embedded dimension $d$ from 5 to 150, and used colors to represent different dimensions. For Exp. 5198, there's accuracy decrease trend with the embedded dimension increasing. But for Exp. 4581 and Exp. 4679, the metrics show more complex situations. Generally lower embedded dimension ($d < 70$, blue to cyan lines) brings better performance in short-term (around 10 step leads), while higher embedded dimension ($d > 100$, yellow to red lines) brings more steady

result. The embedded dimension $(d = 11)$, calculated by Cao' method and dimension we adopted in the manuscript $(d = 100)$, are highlighted with black stars and yellow stars respectively. Noticed that, the metrics when $d = 11$ get a better short-term performance, while the metrics when $d = 100$ get more steady result when the prediction leads increasing. Except for Exp. 5198, the metrics are generally better when $d = 11.$

It indicates that for short-term forecasts, a small embedding dimension is efficient, while also pointing out that for longer-term forecasts, a small embedding dimension may be insufficient. For this interesting phenomenon, we plan to discuss it in our further work.

**Table 2:** Parameters combination results calculated based on Cao's methods (1997). Parameters searching with $\tau = [1, 2 \dots, 10]$ and $d = [1, 2, 3 \dots, 150]$. With the increase of delay time, the embedded dimension become null, which is not listed in this table.

| # No. | Exp. 4679 | | Exp. 4581 | | Exp. 5198 | |
| --- | --- | --- | --- | --- | --- | --- |
| | Delay time ($\tau$) | Embedded Dimension ($d$) | Delay time ($\tau$) | Embedded Dimension ($d$) | Delay time ($\tau$) | Embedded Dimension ($d$) |
| 1 | 1 | 11 | 1 | 11 | 1 | 11 |
| 2 | 2 | 26 | 2 | 26 | 2 | 11 |
| 3 | 3 | 21 | 3 | 36 | 3 | 21 |
| 4 | 4 | 16 | 4 | 26 | 4 | 16 |
| 5 | 5 | 21 | 5 | 26 | 5 | 16 |
| 6 | 6 | 36 | 6 | 21 | 6 | 21 |
| 7 | 7 | 31 | 7 | 16 | 7 | 11 |
| 8 | 8 | 21 | 8 | 26 | 8 | 16 |
| 9 | 9 | 26 | 9 | 36 | 9 | 16 |
| 10 | 10 | 26 | | | 10 | 16 |

[Figure]

[Figure]

[Figure]

**Figure 6.** Embedded dimension (**d**) test for HKAE. The Optimized **d = 11** and **d = 100** adopted in the manuscript are highlighted in black star and yellow star respectively.

**23. Line 197: How are they adapted? Just by trial and error? Or with some consistent procedure?**

**[Response]:**

Thank you for your feedback. They are continuous pathways for model regularization during training (Hoffer et al., 2017).

The weight decay is a skill implemented by subtracting a value proportional to the size of the weights at each weight update.

The gradient clip is a means that limiting the size of the gradient vector before updating the model weights.

They are the commonly used techniques to prevent overfitting in neural network training, and therefore we do not go into detail in the text.

Reference:

[1] Hoffer, E., Hubara, I., & Soudry, D.: Train longer, generalize better: closing the generalization gap in large batch training of neural networks, in: Advances in Neural Information Processing Systems, vol. 30, 2017.

**24. Tab. 1: adopted instead of adapted? These parameters are fixed, not adapting.**

**[Response]:**

Thank you for your valuable suggestion. We used "adopted" in the revised version.

**25. Eq. 14: Is the mean the mean over the whole time series or in the window used to make the forecast? Using the whole time series mean seems wrong to me: why should the forecast tend to the mean of the whole time series when using a specific window that has a subset of the whole time series?**

**[Response]:**

Thank you for your valuable suggestion. The mean is the mean in the forecast window, not the whole time series. In order to express the assessment approach more precisely, we adjusted the formulation of Eqs. 14-15 and add information about sliding window prediction in Figure 4:

For prediction lead steps $j = (1, ..., L)$ we generate forecast window $i = (1, ..., N - K - L + 1)$, we compute the $R_j^2$ and $RMSE_j$ as follows:

$$R_j^2 = 1 - \frac{\sum_{i=0}^{N-K-L+1} (\widetilde{x_i^j} - x_i^j)^2}{\sum_{i=0}^{N-K-L+1} (\overline{x^j} - x_i^j)^2} \tag{14}$$

$$RMSE_j = \sqrt{\frac{1}{N - K - L + 1} \sum_{i=0}^{N-K-L+1} (\widetilde{x_i^j} - x_i^j)^2} \tag{15}$$

Where $j$ is the prediction lead step in the $i - th$ forecast window. $R_j^2$ and $RMSE_j$ represents the $R^2$ and $RMSE$ for $j - th$ lead predictions. $\tilde{x}$, $\bar{x}$, $x$ are respectively the predictions, mean and ground truth of $x$.

**26. Eq. 17: The notation is quite odd. What you name "slip" or "stress" it's actually just the index, so it is a time. When writing "slip_center_index" what is typically understood is that that is the value taken by the slip at the time index given by the index. Furthermore, "the central moment of a slip behaviour" is not clear. What is a "slip behaviour"?**

**[Response]:**

Thank you for your valuable suggestion.

(1) It's true. The indexes in the notation of Eq.17 points to the time nor the stress value at the time index. And $x_{index}$ represents the stress value at certain time index.

(2) The slip behaviour typically refers to the characteristics and patterns of movement observed when two surfaces, often on a fault or an interface, slide past one another. It is usually accompanied by stress relief. And "the central moment of a slip behaviour" represents the denotes the closest points during a slip event to the average value of the shear stress (Gualandi et al., 2023).

(3) We simplified the notations and changed the text in Eqs.16-17 and Figure 6 in the revised version:

$$\Delta T_i = \Delta t * (t_{i+1}^{cen} - t_i^{cen}) \tag{16}$$

$$t_i^{cen} = t_i^{top} + argmin(|x(t_i^{top}), \dots, x(t_{i+1}^{bot})|) \tag{17}$$

[Figure]

**Figure 7.** Revised Fig. 6 in manuscript.

**27. Eq. 18: The fraction and parenthesis are likely wrong: check the consistency of your equation.**

**[Response]:**

Thank you for your feedback. We corrected Eq. 18 in the revised version:

$$R_{slip_j}^2 = 1 - \frac{\sum_{i=0}^{N_{slip}} (\widetilde{\Delta T_i^j} - \Delta T_i^j)^2}{\sum_{i=0}^{N_{slip}} (\overline{\Delta T^j} - \Delta T_i^j)^2} \tag{18}$$

Where $j$ is the prediction lead step, $i$ is the number of slips in the $j^{th}$ predictions. $\widetilde{\Delta T}$, $\overline{\Delta T}$, $\Delta T$ are respectively the predictions, mean and ground truth of $\Delta T$.

**Section 3: Results**

**1. Lines 237-239: SVD is not applied to validate the effectiveness of the dynamic modelling.**

**[Response]:**

Thank you for your valuable suggestion. We think that SVD is a necessary tool to test the validity of the dynamic

modeling process. It verifies the degree of information retention before and after encoding (nonlinear mapping) in the HKAE framework by extracting the main components. It is manifested in the fact that the system trajectories are still preserved, which can be clearly observed in Figure 7b-c, especially for ODE simulation. We believe that SVD is a necessary part of effectiveness validation, although it does not provide a direct indication. To avoid ambiguity, we modify the statement to:

"We further check the data status during the model procedure. We employ Singular Value Decomposition (SVD) to confirm the trajectory of system, and Eigen Decomposition (ED) to learned operator."

**2.  Lines 243-244: Is it really quasi-periodic?**

**[Response]:**

Thank you for your feedback. Regarding the statement of "quasi-periodic", we have included in the Response 4 in Section 2, mainly by referring to existing work and performing time-frequency domain analysis.

**3.  Lines 249-253: The wording is very cryptic. What do you mean by "a single observation dimension"? Furthermore, a dynamical system will follow a trajectory that defines the attractor. How is it possible to have an unstable stick-slip system where the data does not exactly follow the attractor trajectory during the experiment?**

**[Response]:**

Thank you for your feedback.

(1) The term "a single observation dimension" refers to the fact that for a dynamical system such as a laboratory earthquake, we used only a **single observation of shear stress** in our experiments. According to previous studies, this observation is a good indicator of the state change of the laboratory seismic system (Gualandi et al., 2023).

(2) The use of term "unstable" is inaccurate and we replaced "unstable" with "**metastable**" in the revised version. We gain the conclusion of a laboratory earthquake system as a "meta-stable system" from Jasperson' work (2021).

Reference:

[1]  Gualandi, A., Faranda, D., Marone, C., Cocco, M., Mengaldo, G., and Bendick, R.: Deterministic and stochastic chaos characterize laboratory earthquakes, Earth and Planetary Science Letters, https://doi.org/10.1016/j.epsl.2023.117995, 2022.

[2]  Jasperson, H., Bolton, DavidC., Johnson, PaulA., Guyer, RobertA., Marone, C., and Hoop, MaartenV. de: Attention network forecasts time-to-failure in laboratory shear experiments, Journal of Geophysical Research: Solid Earth, https://doi.org/10.1029/2021JB022195, 2021.

**4.  Line 257: Experiment p5198 does not have a stable mode at 9 s.**

**[Response]:**

Thank you for your feedback. The mode for Exp. 5198 is actually around 6.39s (Table 2). We modified it in the revised version.

**5.  Lines 259-260: Why a zero-frequency mode with low amplitude is representative of a strong time-**

**varying component? What does it mean "strong time-varying component"?**

[Response]:

Thank you for your feedback. Our original statement might be ambiguous and we corrected it in the revised version as follows:

"…but the amplitude of the zero-frequency mode in Exp. 4679 with slip switch is much lower, which means the static component is a small percentage of the shear stress variation comparing to the other 2 experiments. This may indicate that when a laboratory earthquake system is in a state of alternating fast and slow slips, the time-varying dynamics of the system itself will be significantly enhanced."

**6. Line 278: Why only two experiment settings?**

[Response]:

Thank you for your feedback. The experiment settings here represent the setting of the prediction task above ("historical 10s to predict future 3s; historical 20s to predict future 10s"), not the actual shear experiment shown in Figure 1a. To avoid confusion, we have claimed the details in the revised version:

"We compare our results under 2 prediction settings with 3 laboratory stick-slip datasets and 3 comparative deep learning time series prediction methods."

**7. Line 284: You claim that the HKAE results are clearly better when using the DeltaT evaluation metric. OK, it is better, but the R2 is most of the time negative, and, in the best situation, slightly positive. This seems to be a very poor result for all the architectures. Even if for the HKAE it is better, it is still a poor result.**

[Response]:

Thank you for your feedback.

First of all, we recognize that the performance of statistical metrics isn't good. But objectively speaking, it is undeniable that autoregressive prediction of stress changes in laboratory earthquake systems over a certain time window is a difficult task. The following figure shows the prediction evaluations from the Laurenti's work (2022).

[Figure]

[Figure]

**Figure 8.** Fig. 6, 7 from Laurenti et al., 2022

From 2 figures above, we find that the future predictions of shear stress are not that accurate as we hoped. The R-square also keeps relatively low, even negative for the Exp. 4581, which also happened in our work. On the one hand the sudden and rapid release of stress can be regarded as a "switch" of system, which is more difficult to predict; on the other hand, the statistical criteria are also sensitive to the data with rapid changes. This is one of the reasons that we developed new evaluation metrics (ΔT evaluation) that consider the timing of event.

Note that, we adopt different prediction evaluation strategies with Laurenti. We used an evaluation method commonly used in fields such as weather forecasting, where the results are evaluated across all windows at different prediction leads, to fairly assess the model's performance against an increasing number of prediction leads. However, in Laurenti's case it is a direct averaging of the content prediction results for different prediction windows, so her statistical score performance will differ from ours.

Reference:

[1] Laurenti, L., Tinti, E., Galasso, F., Franco, L., and Marone, C.: Deep learning for laboratory earthquake prediction and autoregressive forecasting of fault zone stress, Earth and Planetary Science Letters, https://doi.org/10.1016/j.epsl.2022.117825,2022.

8. **Line 285-287: The sentence is unclear. It is surprising that the RMSE and R2 metrics are worse for the HKAE in the first few forecasted steps. This would suggest that for short term predictions LSTM and TCN are better. But the dynamics that you are extracting should degrade with longer lead prediction times. Something that instead is not really observed.**

[Response]:

Thank you for your feedback.

(1) We further investigated the short-term prediction performance of LSTM and HKAE, taking Exp.4581 as example. We illustrated 3s lead predictions during different stress release phase, and marked the 10-step predictions in gray whereas HKAE has poorer $R^2$ and RMSE. We believe that the main reason for the good performance of LSTM in the short-term prediction is that, thanks to the ability of LSTM in modeling short-term time-series dependence, more accurate results can be obtained in the stress-raise phase. For HKAE, it lacks in numerical precision, and at the same time, there will be slight perturbations, which leads to a lower final score than that of LSTM. However, we can also find that the prediction of LSTM in the stage of stress

release is much worse than that of HKAE.

And since the percentage of the stress-release phase is larger than that of the stress-release in our data, it ultimately leads to a better result for LSTM when averaged over a sliding window. After 1s, the advantage of HKAE for dynamics modeling is demonstrated, and the predictions in the stress-release phase converge to the true value, thus appearing to essentially equal as well as exceed the LSTM.

On the other hand, from delay embedding test of HKAE, it's able for HKAE to generate competitive short-term prediction results with a lower embedding dimension (Detail in Response 21 in Sec. 2). In general, methods like LSTM do have an advantage in modeling short-term temporal dependencies, but when we focus on the scenario-specific predictions (e.g., stress-release phases) as well as longer-term dependencies, HKAE's performance is better.

We add the Figure 9 in revised version to show the modeling ability of HKAE during stress release phase.

[Figure]

**Figure 9.** Historical 10s predict future 3s experiment details extension, more sections are added to show prediction difference between HKAE and LSTM. The first 10 steps (1s) lead predictions are masked in gray patches. Using $R^2$ and RMSE as evaluation metrics, and predictions with higher metrics are highlighted in red.

(2) Regarding the attenuation of prediction accuracy with lead prediction steps, theoretically such a phenomenon should occur, and in fact it occurs in all 6 cases (2 prediction settings and 3 experimental data) that we evaluated (manifested by the decrease of $R^2$ with lead and the increase of RMSE with lead). The HKAE method tends to show a weaker decay trend, which reflects the effectiveness of HKAE in capturing dynamic information. The dynamic modes in Figure.11 b, on the other hand, is the evolution of the individual eigenmodes over time after approximating the Koopman operator eigen decomposition. The amplitude of the first two modes is less than 1, and it should be essentially stable over time (Detail in Response 14 in Sec.3). And it is because of the existence of such stable modes that it is possible to get still more stable predictions (i.e., slower decay of accuracy) over longer lead times.

(3) Regarding to your comments, we revised the text as follows for clearer statements:

"But from the results of our slip interval ΔT modelling metrics…the traditional deep learning methods appear to have poorer future estimates of future event periods than the HKAE. The HKAE holds a higher initial miss but get more steady result as prediction leads increases. Specifically, for HKAE it is a slower increase in RMSE with predicted steps, slower raise in $R^2$, and a steadier result in $R^2$ of event periods. We attribute this more robust future estimation of to HKAE's modeling of the dynamics of the system as it evolves."

9. **Lines 289-294: Which metric? You tested more than one. The setting 20-10 s is shown in Fig. 8b, not 8a.**

   **Finally, how can you claim that the lead prediction can cover 1-2 complete cycles? When we look at Fig. 8b, the DeltaT metric shows that many points are not on the diagonal, meaning that the prediction is wrong in many cases regarding the timing of the events. This class misrepresentation (DeltaT small or large are the two classes) indicates that your lead prediction cannot cover multiple cycles.**

[Response]:

Thank you for your feedback.

(1) We have corrected unspecified metrics and incorrectly labeled figures in the revised version:

   In addition, we can find that in the lead prediction of the three experiments, the $R^2$ metric will decay more rapidly in accuracy before 1s lead, while the decay slows down after 1s, especially for the traditional machine learning method. Under the "20s-10s" experimental setting (Figure 8b), we believe that the lead prediction can cover 1-2 complete seismic cycles (Laurenti et al., 2022), at which point the statistical period of the slip interval will be longer and more representative. Compared with the experimental settings of "10s-3s" (Figure 8a), predicting the results for the next 10s, the slip interval scores of the HKAE predictions get higher than other methods, which reflects the superiority of HKAE in modeling seismic dynamics.

(2) First of all, we want to claim is that the original text "the lead prediction can cover 1-2 complete seismic cycles" in the manuscript is for the setting of the prediction window is long enough to cover 1~2 cycles (1 cycle for Exp. 4581, 2 cycles for Exp. 4679 and Expo. 5198, as shown in following Figure 7).

   And the reason that the $\Delta T$ index exhibits both large and small values (for Exp. 4679) is because there are alternating fast and slow stresses in Exp. 4679, so there will be two large categories of $\Delta T$ indicating different types of slip behavior. Our test data for Exp. 4679 reveals the existence of two major groups of $\Delta T$, too (Figure 1c in Sec.1 Response 4).

[Figure]

**Figure 10.** Illustration of lead prediction horizons for 3 experimental data.

Legend: ▮ Prediction setting 1: 3s prediction window  ▮ Prediction setting 2: 10s prediction window

(3) Being able to accurately predict 1-2 complete cycles in the future is difficult. We admit that the performance of HKAE is not good enough, but what we want to emphasize here is that it is possible to predict future event cycles more accurately with HKAE than with LSTM represented machine learning methods (for statistical methods, we test in Sec.3 Response 13). We think this is already an improvement.

(4) To avoid ambiguity, we modified statements here:

"Under the "20s-10s" experimental setting (Figure 8a), which the horizon can cover 1-2 complete seismic cycles (Laurenti et al., 2022), at which point the statistical period of the slip interval will be longer and more representative."

**10. Line 296: If the slip events were periodic, it would be very easy to predict them, and you should have zero error. The events are not periodic.**

**[Response]:**

Thank you for your feedback.

We agree that slip events are not periodic. Here we use "periodicity" to represent the certain degree of regularity, which does not mean that it is entirely periodic. To avoid misunderstanding, we replace "periodicity" with "cyclicity".

**11. Figure 8: In the main text you start from experiment 5198, and this should be the first to appear in the figure (i.e., on the left).**

**Furthermore, you have also shown ODE simulations earlier on. You should show the results on the simulations as well. They are periodic, so you should be able to recover them perfectly. If not, this would cast some doubts on the procedure.**

**[Response]:**

Thank you for your valuable suggestions.

We adjusted the order of experiment in the figure, and add the ODE experiment in the Supplementary Information.

(1) Our purpose of using ODE simulation data is that its performance in reconstructing the attractor subspace

is more homogeneous to facilitate readers' understanding of Figure7. Thus, the results of ODE are not
shown in the results part.

(2) Taking your suggestions into account, we have supplemented the experimental results of the ODE
simulations below, and we can find that the ODE simulations of HKAE keeps a high accuracy, which
further proves the validity of the HKAE modeling.

[Figure]

**Figure 11.** Historical 10s predict future 3s using ODE simulations.

12. **Lines 318-322: There is confusion on what you mean by underestimation. For example, if you say
that the minimum value is underestimated, it means that the predicted shear stress is smaller than
the actual minimum shear stress. But the figure shows that the predicted minimum shear stress is
higher. What you mean is that the stress drop is underestimated. Please be consistent and describe
properly the results.**

[Response]:

Thank you for your suggestions.

What we were trying to convey is that the absolute value of the stress drop was underestimated, as evidenced by
the fact that the minimum value of the predicted stress was higher than the minimum value of the actual stress,
and the statement has been updated in the revised version:

"Notably, Exp. 4679 and Exp. 5198, which include slow slips, provide more accurate predictions of maximum
stress values during slips, though they tend to overestimate the minimum value after stress release. Exp. 4581,
on the other hand, predicts a narrower range of stress variation, with a lower maximum and a higher minimum
stress value".

13. **Line 326: "moment of" -> "instant for".**

[Response]:

Thank you for your valuable suggestion. We have modified it in the revised version.

14. **Line 343-346: I disagree with the way you are reading the figure. In Exp. 4679 LSTM clearly has a
better RMSE for most of the lead time between 3 and 10 s (non-grey area). It is not even clear how
you are drawing those lines. If you look at just the points (and not the interpolated line, which is not
an actual result), LSTM is in fact comparable to HKAE until 4.5 s, and then always better or with
same results. For Exp. 4581 the RMSE is basically the same for all algorithms. Finally, for Exp.5198
the RMSE of LSTM is the best until about 7 s, where it becomes like the one of HKAE, and only**

**slightly worse after 9 s. But what does it mean to be able to have a slightly better forecast at 9- 10 s lead time? Based on your results, if I had to pick an algorithm to make earthquake forecasts, I would use LSTM: it clearly outperforms HKAE in the forecast of the immediate next steps, which is what matters the most. For long-term forecasts (like 9-10 s, which is a full cycle ahead), one can very likely use purely statistical methods and obtain similar results.**

Thank you for your comments. To avoid ambiguity, we have decided to remove the Section 3.3 in the revised version, and discuss the out-horizon prediction in our further work. We provide the following analysis for response.

(1) Considering the ambiguity caused by this section, we plan to temporarily remove section 3.3 from the revised manuscript. Additional experiments for this section are as follows, which we plan to discuss in further work.

(2) During our experiments, we realized that we can't use simple metrics to evaluate the estimated results of laboratory slip, because we don't just focus on the numerical change of shear stress, but also on other points, such as when the stress is released and how long it takes for the release to start recovering. You evaluated our results mainly from RMSE from your comments. However, in reality, we arrived at our conclusion by combining the three scores as well as the actual predicted results. In the modified Figure 10, we have labeled HKAE's dominant lead steps for different metrics (Translucent red vertical lines).

For Exp. 4581 (Laurenti et al., 2022), which is the most difficult to predict, HKAE has better performance in $R^2$, $R^2$ of $\Delta T$ in leading steps after 4s, although its RMSE results do not differ much from other methods. From the actual leading prediction results, HKAE is better than LSTM in both stability and accuracy of prediction results.

For Exp. 4679 and Exp. 5198, HKAE is not as bright as Exp. 4581 in $R^2$ and RMSE, but from the results of $R^2$ of $\Delta T$, HKAE still has good performance in more leading steps. We also note that for Exp. 4679, HKAE final prediction results will have larger numerical fluctuations compared to LSTM. We performed a short-time Fourier transform on the stress data of Exp. 4679 and found that its frequency components in the test set portion changed considerably, leading to the fact that it is more difficult for HKAE to stably estimate systematic variations with the global operator modeled from the training set, whereas LSTM is able to obtain relatively better results through the ability of short time-series dependent modeling.

**Out horizon lead predictions evaluation for Exp. 4581**

**LSTM Lead predictions for Exp. 4581**

**HKAE Lead predictions for Exp. 4581**

**Out horizon lead predictions evaluation for Exp. 4679**

**LSTM Lead predictions for Exp. 4679**

**HKAE Lead predictions for Exp. 4679**

**Out horizon lead predictions evaluation for Exp. 5198**

**LSTM Lead predictions for Exp. 5198**

**HKAE Lead predictions for Exp. 5981**

**Figure 12.** Modified Fig. 10 in manuscript.

[Figure]

**Figure 13.** Short-Time Fourier Transform for Exp. 4679. Red box indicates the changes in frequency components in the test set.

For Exp. 5198, we believe HKAE generates comparable to other methods in terms of final prediction results (Figure 14). LSTM gets more accurate results in the initial leads (0-5s), while the HKAE results are comparable in terms of prediction results. For the better results predicted by HKAE in the later prediction leads, we believe is highly correlated with its ability to model the system dynamics. We further extend the prediction horizon to 20 s and find this characteristic more clearly manifested (Figure 15).

[Figure]

**Figure 14.** predictions of 10s out horizon experiment for Exp. 5198.

[Figure]

**Figure 15.** Evaluations and predictions of 20s out horizon experiment for Exp. 5198. HKAE shows more robust performance over a longer predict horizon.

(3)  We supplemented our experiments with ARIMA, the commonly used statistical method in time series prediction, under the same prediction settings (Figure 12), and we can find that it achieves competitive results to LSTM, HKAE, and other methods under near-instantaneous prediction (prediction lead < 0.5s), but the accuracy decays more severely as the prediction lead increases. We think that they may not work as well as you would expect when discussing longer period prediction scenario.

[Figure]

**Figure 16.** Historical 10s predict future 3s experiment evaluations, ARIMA experiments are added.

15. **Lines 369-370: The two "stable" dynamics clearly show an amplification over time, which makes them not as stable as you depict them.**

    **From Fig. 11b it is not clear what they represent. What do you mean with "input"? These are modes obtained after processing the input data in some way.**

[Response]:

Thank you for your valuable suggestion. We provide the following analysis for response.

(1) Firstly, the stability of the dynamic mode is determined by the amplitude of the eigenvalue (Brunton et al., 2022). The dynamic modes can be considered as stable mode when its norm of eigenvalue keeps a narrow threshold ($1\pm0.001$ in Avila and Mezic, 2020). The former 2 eigenvalue are 1.0007324 and 1.001387, which are very close to the threshold. We think it indicates that systems with alternating fast-slow slips are more unstable than systems with fast and slow rupture.

(2) The "input" here is the latent variables after HKAE encoding. We modified the legend for clear representation.

Reference:

[1] Brunton, S. L., Budišić, M., Kaiser, E., and Kutz, J. N.: Modern Koopman Theory for Dynamical Systems., SIAM Review, 229–340, https://doi.org/10.1137/21m1401243, 2022.

[2] Avila, AllanM. and Mezic, I.: Data-driven analysis and forecasting of highway traffic dynamics., Nature Communications, https://doi.org/10.1038/s41467-020-15582-5, 2020.

16. **Lines 387-388: This is not true. For short-term predictions HKAE seems to perform poorly compared to other algorithms. If we look at either Fig. 8 or Fig. 10, we see that for the first few time-steps of forecast the HKAE is consistently not the best, with LSTM being often the best, followed by TCN.**

[Response]:

Thank you for your valuable suggestions.

From the test of embedded dimensions, we found that HKAE under small embedded dimension is able to get comparable results to LSTM in the short-term prediction situation. Considering the emphasis in this paper on HKAE's ability to model system dynamics, we have adjusted the statements in the revised version:

"Considering recent success on dynamic system embedded real-word earthquake applications (Tong et al., 2023) and dynamic modeling skills of HKAE, it would be of potential to modeling seismicity over a certain period for reconstruction or prediction based on HKAE."

**17. Lines 392-394: This is not true. On many occasions HKAE was not showing superior performance in terms of prediction accuracy.**

[Response]:

Thank you for your valuable suggestions. Original statement may be misleading and we have more objectively expressed the modeling results and advantages of HKAE in the revised version:

"Drawing upon delay embedding and Koopman theories, we have proposed a dynamic informed machine learning method, the Hankel Koopman Auto-Encoder (HKAE), to achieve future predictions of complex laboratory earthquake systems. The model reconstructs the phase space of the slip system through stress variations and executes the system's dynamic evolution by approximated Koopman operator. Through experiments with varying slip characteristics, it is found that HKAE achieves comparable results to deep learning methods in predicting the future states of the slip system, particularly excelling in stress estimation during the stress release phase and in modeling event intervals over long-term. Due to HKAE being a modeling approach driven by dynamical systems theory, it provides insights into the system's dynamics that are inaccessible to traditional deep learning methods. HKAE indicates the presence of stable dynamic modes within the laboratory slip system, with the stability of the alternating fast-slow slip system being weaker than that of the fast or slow rupture systems alone. HKAE's modeling performance from single observation and interpretability holds significant potential for monitoring fault behavior and future activity assessment under real-world observational constraints."

**Reply to Review for "A dynamic informed deep learning method for future estimation of laboratory stick- slip" by Yue et al.**

**[Review]:**

This manuscript explores a novel model that integrates dynamic systems theory with the nonlinear fitting capabilities of deep learning. The HKAE model utilizes the Koopman operator and an autoencoder framework to reconstruct the dynamic behavior of laboratory slip systems, with a specific emphasis on shear stress variations. Through a synthesis of theoretical analysis and experimental validation, the HKAE model showcases exceptional performance in predicting complex nonlinear systems. This study underscores the HKAE model's advantages in handling intricate nonlinear dynamic systems and proposes promising directions for future research and applications. The following issues could enhance the manuscript's quality.

In Lines 70-71, the statement "our model envisions the future prediction as the continuous evolution of laboratory fault slip systems" lacks clarity and requires a detailed explanation.

In Section 2.1, simple simulation data was utilized. Did the model explore more intricate numerical simulations, and what were the specific outcomes? More details about the simulation should be given. Where is the laboratory data coming from? At least the references should be provided.

In Section 2.2, please elucidate the advantages of Koopman theory and delay embedding theory in managing the intricate dynamics of laboratory fault slip systems.

Line 97, "we're" should be "we are". No contraction is allowed in formal English writing. Line 340, as well, among others.

In Section 2.3, how do the functions of these three model modules impact the model's performance?

Line 305, "to shows" should be "to show". The authors should carefully check the whole paper regarding typos.

In Figure 9, RMSE and R2 should be included.

In Line 339, the model exhibits subpar performance in Exp. 4581. Is the model particularly sensitive to specific types of data?

In Section 4, what is the interpretability of the HKAE model? Are there specialized methods or techniques for elucidating the model's predictions?

In Section 5, the HKAE model has demonstrated robust predictive capabilities. What are the potential directions for future research?

The fonts in some figures are extremely large, while in others are too small to read. I highly recommend that the authors re-draw most figures to enhance their quality.

**[Response]:**

Thank you so much for your valuable comments, which are very helpful for improving our manuscript.

In response to your review comments, we will make the following responses:

(Black bold text: Reviewers' comments; Purple text: Our responses; Red text: changes in manuscript)

1. **In Lines 70-71, the statement "our model envisions the future prediction as the continuous evolution of laboratory fault slip systems" lacks clarity and requires a detailed explanation.**

**[Response]:**

Thank you for your comments. Here we want to emphasize that we do not define the problem as a time series forecast from a statistical point of view, but rather from a dynamical system perspective. Specifically, we take the shear stress time series as one of the observations of a laboratory slip system, and infer changes in the future state of the system through methods inspired by dynamical systems theory. To this end, we reconstruct the phase space of the system using delayed embedding theory and linearize its complex dynamics using Koopman theory to perform future inference and dynamical analysis.

To explain more clearly, we changed the statements to: "Instead of defining the problem as a statistical time series forecast task, we take the shear stress time series as one of the observations of a laboratory slip system, and infer changes in the future state of the system through methods inspired by dynamical systems theory. Generally, we reconstruct the phase space of the system using delayed embedding theory and linearize its complex dynamics using Koopman theory to perform future inference and further dynamical analysis."

2. **In Section 2.1, simple simulation data was utilized. Did the model explore more intricate numerical simulations, and what were the specific outcomes? More details about the simulation should be given. Where is the laboratory data coming from? At least the references should be provided.**

**[Response]:**

Thank you for your comments. The simulation data is conducted with code from Gualandi's work (2023) with these following equations:

$$\frac{\partial x}{\partial t} = \frac{e^x[(\beta_1 - 1)x(1 + \lambda u) + y - u] + \kappa\left(\frac{y_0}{v_*} - e^x\right) - \frac{\partial u}{\partial t}\frac{1 + \lambda y}{1 + \lambda u}}{1 + \lambda u + v e^x}$$

$$\frac{\partial y}{\partial t} = \kappa(\frac{v_0}{v_*} - e^x) - v e^x \frac{\partial x}{\partial t}$$

$$\frac{\partial z}{\partial t} = -\rho e^x(\beta_2 x + z)$$

$$\frac{\partial u}{\partial t} = -\alpha - \gamma u + \frac{\partial z}{\partial t}$$

Where $[x, y, z, u] = [\ln\left(\frac{v}{v_*}\right), \frac{\tau_f - \tau_0}{a\sigma_{n0}}, \frac{1}{\lambda\beta\sigma_{n0}}(\phi - \phi_0), -\frac{1}{\lambda}\frac{p}{\sigma_{n0}}]$ represents the system state. Here we set the normal stress initial as $\tau_{n0} = 17.003\ Mpa$ and state vector initial as $[x_0, y_0, z_0, u_0] = [0.05, 0.0, 0.0, 0.0]$ to generate fast-slow-switching slips as simulation examples. We supplied the equations and parameters used above during simulation in the revised version.

We mainly focus on the actual laboratory data, while the simulation data is used as a conceptual and pure sample, to make the further dynamic analysis easier to understand. More complex numerical simulation data, e.g., weak cyclicity data (Wang et al., 2021) we consider to discuss in our further work.

The laboratory data comes from Laurenti's work (2022), the original source is http://psudata.s3-website.us-east-

2.amazonaws.com/. We have annotated in Section 2.1 as "The experimental data from the biaxial shear equipment comes from the PSU laboratory (Laurenti et al., 2022)" and in Code and data availability.

Reference:

[1] Gualandi, A., Faranda, D., Marone, C., Cocco, M., Mengaldo, G., and Bendick, R.: Deterministic and stochastic chaos characterize laboratory earthquakes, Earth and Planetary Science Letters, https://doi.org/10.1016/j.epsl.2023.117995, 2022

[2] Laurenti, L., Tinti, E., Galasso, F., Franco, L., and Marone, C.: Deep learning for laboratory earthquake prediction and autoregressive forecasting of fault zone stress, Earth and Planetary Science Letters, https://doi.org/10.1016/j.epsl.2022.117825,2022

[3] Wang, K., Johnson, C. W., Bennett, K. C., and Johnson, P. A.: Predicting fault slip via transfer learning., Nature Communications, https://doi.org/10.1038/s41467-021-27553-5, 2021

3. **In Section 2.2, please elucidate the advantages of Koopman theory and delay embedding theory in managing the intricate dynamics of laboratory fault slip systems.**

**[Response]:**

Thank you for your suggestions. We have further elucidated the advantages in the revised version.

(1) The advantage of delay embedding theory is that using only the shear stress observations to reconstruct the phase space of laboratory system. Considering the limited observation of fault system, it's potential to use stress series (or stress proxy like displacement observations) to recover the system behaviours.

(2) The advantage of Koopman theory is to linearize the dynamics from complex laboratory fault system, then support the analysis of system behavior using linear analysis tools (e.g., singular value decomposition), which will offer the interpretability and insights from the dynamical system perspective.

4. **Line 97, "we're" should be "we are". No contraction is allowed in formal English writing. Line 340, as well, among others.**

**[Response]:**

Thank you for your suggestions. We have adjusted these informal expressions in revised version.

5. **In Section 2.3, how do the functions of these three model modules impact the model's performance?**

**[Response]:**

Thank you for your comments. Since the HKAE is designed inspired by dynamical systems theory, the modules are more closely connected, making it more difficult to discuss a thorough separation. We discuss the roles of the modules theoretically and perform alternative ablation experiments.

(1) The delay embedding module reconstructs the phase space of system. Without delayed embedding to provide embedding coordinates, the latter two modules are unable to construct stable mappings and approximate Koopman operators from univariate data alone. Without delayed embedding to provide embedding coordinates, the latter two modules are hard to construct stable mappings and approximate Koopman operators from univariate data alone (Brunton et al., 2021).

(2) The mapping learning module performs the necessary nonlinear mapping, which, if removed, becomes a linear moving-average-like method. We've tested the ARIMA for the most challenging fast-slow-switching

slips and the results shows that the linear methods do not perform well (Figure 1).

[Figure]

**Figure 1.** Ablation study using historical 10s to predict future 3s. Green line illustrates the performance of ARIMA, representing linear methods without nonlinear mapping.

(3)  The Koopman operator achieve linear evolution of system in the latent space, and offer dynamic interpretability. We change the Koopman operator into a linear layer of equal size but with bias, and test the "10-3s" predictions for the most challenging fast-slow-switching slips. The statistical metrics illustrate that HKAE performs better than the model without Koopman evolution module. More importantly, removing this module makes it impossible to analyze the dynamical patterns of the system.

[Figure]

**Figure 2.** Ablation study using historical 10s to predict future 3s. Blue line represents the HKAE model whose operator in Koopman Evolution module is replaced by a normal linear layer with activation.

Reference:

[1]  Brunton, S. L., Budišić, M., Kaiser, E., and Kutz, J. N.: Modern Koopman Theory for Dynamical Systems., SIAM Review, 229–340, https://doi.org/10.1137/21m1401243, 2022.

6.  **Line 305, "to shows" should be "to show". The authors should carefully check the whole paper regarding typos.**

**[Response]:**

Thank you for your suggestions. We have checked the typos in revised version.

7.  **In Figure 9, RMSE and R2 should be included.**

**[Response]:**

Thank you for your suggestions. We have added statistical metrics in the figure.

8. **In Line 339, the model exhibits subpar performance in Exp. 4581. Is the model particularly sensitive to specific types of data?**

**[Response]:**

Thank you for your comments. Theoretically, the model is designed entirely from dynamical systems theory, which is a generalized structure and therefore not sensitive to specific data.

Theoretically, the model is designed entirely from dynamical systems theory, which is a generalized structure and therefore not sensitive to specific data. Objectively, HKAE's results on the Exp.4581 are poor because of its high prediction difficulty, but compared to other methods, HKAE actually has a more competitive edge (Figure 9 in manuscript). We infer the true dimension of the laboratory slip system will also have a certain impact on the performance of HKAE. We analyze slow slip experiments (Exp. 5198) that are statistically average in terms of competitiveness in performance, and find that they are significantly better able to achieve better performance in low embedding dimensions (Figure 3).

Taking into account your suggestions for more complex simulation data, we will further discuss the sensitivity of HKAE to data with different characteristics in our follow-up work.

[Figure]

**Figure 3.** Embedded dimension test for slow slip system. Black line indicates the embedded dimension calculated by Cao et al., while yellow line represents the embedded dimension used in the manuscript.

9. **In Section 4, what is the interpretability of the HKAE model? Are there specialized methods or techniques for elucidating the model's predictions?**

**[Response]:**

Thank you for your comments. To enhance the expression of HKAE interpretability, we have provided a more detailed explanation in the revised version.

The interpretability of HKAE comes from the delay embedding theory and Koopman theory. The encoder of HKAE mapped the phase space of input shear stress into a latent space, where we think the evolution of the system is linear. And the linear dynamics are controlled by the matrix-like approximate Koopman operator. Thus,

the characteristic of system can be discussed. The Similar interpretations can be found in works by Lusch et al. (2018), Azencot et al. (2020) and Ouala et al. (2023).

The amplitude and of dynamic are shown in Figure 7 and introduced in Section 3.1. Section 4 further discuss the dynamics evolution characteristics (Figure 11a). Then Figure 11c and 11d discussed the latent variables evolution under the approximate koopman operator, which indicates that the HKAE can obtain components that can be linearly modeled in the latent space (the first 7 subplots), but there will still be some components that are difficult to describe with linear dynamics. There could be two reasons for this result: one is the limitation of modeling due to insufficient observations, where some of the system's dynamics need to be explained as nonlinear forcing (Brunton et al., 2017); the other is the complexity of the system's dynamic characteristics. We've been considering discuss it in our further work.

Reference:

[1]  Lusch, B., Kutz, J. N., and Brunton, S. L.: Deep learning for universal linear embeddings of nonlinear dynamics., Nature Communications, https://doi.org/10.1038/s41467-018-07210-0, 2018

[2]  Azencot, O., Erichson, N. B., Lin, V., and Mahoney, MichaelW.: Forecasting Sequential Data using Consistent Koopman Autoencoders, International Conference on Machine Learning. PMLR, 2020.

[3]  Bounded nonlinear forecasts of partially observed geophysical systems with physics-constrained deep learning

**10. In Section 5, the HKAE model has demonstrated robust predictive capabilities. What are the potential directions for future research?**

**[Response]:**

Thank you for your suggestion. We'd like to further discuss 2 directions:

(1) One of the potential directions is considering the time-vary dynamics in the system. There're time-vary dynamics since we analyze the time-frequency analysis of the shear stress of Exp. 4679. It' s challenging for HKAE to model the time-vary dynamics since the Koopman operator is trained to become a "global operator" (Brunton et al., 2022; Liu et al., 2023). The global operator is less likely to make accurate predictions when there are large changes in the dynamical features.

(2) The second one is to optimize the modeling framework in the presence of only a single observation to obtain more accurate estimates of the system dynamics. For example, considering the laboratory system as a forced system (Brunton et al., 2017), whose force may be processed acoustic emissions.

Reference:

[1]  Brunton, S. L., Brunton, B. W., Proctor, J. L., Kaiser, E., and Kutz, J. N.: Chaos as an Intermittently Forced Linear System, Nature Communications, https://doi.org/10.1038/s41467-017-00030-8, 2017.

[2]  Brunton, S. L., Budišić, M., Kaiser, E., and Kutz, J. N.: Modern Koopman Theory for Dynamical Systems., SIAM Review, 229–340, https://doi.org/10.1137/21m1401243, 2022.

[3]  Liu, Y., Li, C., Wang, J., and Long, M.: Koopa: Learning Non-stationary Time Series Dynamics with Koopman Predictors, arXiv: Learning, 2023.

**11. The fonts in some figures are extremely large, while in others are too small to read. I highly recommend that the authors re-draw most figures to enhance their quality.**

[Response]:

Thank you for your suggestion. We have adjusted the figures and re-drawn the figures in revised version.

---

## Author Response (AR2)

**Reply to Review for "A dynamic informed deep learning method for future estimation of laboratory stick- slip" by Yue et al.**

**[Review]:**

Dear Editor,

The manuscript forecasts shear stress in a double-shear stick-slip experiment using an MLP-based autoencoder enhanced by a Koopman operator and a delay-embedded input. The authors chose this ML model due to its superior performance in capturing periodic phenomena, making it suitable for shear stress forecasting. I appreciate that two reviewers have provided insightful feedback and that the authors have made an effort to address all comments. However, rather than merely responding to the reviewers' remarks, I suggest integrating these valuable points and discussions into the main text.

My primary concerns are as follows:

1. As Reviewer #1 pointed out, the authors tend to overstate their results, despite (a) clear instances of underestimation or overestimation in multiple cycles, and (b) performance that is comparable to other models, such as LSTM, in many cases. Additionally, the interpretability argument presented by the authors is a general observation about the system's dynamics rather than a concrete improvement in forecasting accuracy or model generalization.

2. The manuscript lacks a discussion on the model's applicability to other parameters, such as time to failure, as well as its relevance to real-world field studies. It is evident that stress values are only accessible in laboratory experiments and cannot be directly extrapolated to tectonic earthquakes. Therefore, what are the potential applications of this model in such scenarios? Furthermore, while shear stress exhibits a highly periodic behavior in double-shear tests, rough fault stick-slip experiments (Goebel et al., 2012; Dresen et al., 2020) have demonstrated that roughness evolution significantly influences stress cycles. The authors should include a discussion on this aspect.

Additionally, I have a few minor comments:

1. Figures 8 and 10 contain numerous plots. For instance, in Figure 10, if the first two columns represent different views of the same plot, they should be visually linked using lines or another connecting element. And use some kind of numbering gor them.

2. Figure 10: What does a negative R2 value indicate?

**[Response]:**

Thank you so much for your valuable comments, which are very helpful for improving our manuscript.

In response to your review comments, we will make the following responses:

(Black bold text: Reviewers' comments; Purple text: Our responses; Red text: changes in manuscript)

1. **However, rather than merely responding to the reviewers' remarks, I suggest integrating these valuable points and discussions into the main text.**

**[Response]:**

Thank you for your well-founded comments. In our previous discussions with reviewers, we were deeply inspired. We have compiled valuable insights from those discussions and incorporated them into the revised manuscript, primarily including:

1. Time-frequency analysis of stress variations across different slip behaviors, slip interval statistics, and their relationship with prediction performance.

2. Performance differences of HKAE under varying slip modes and prediction horizons, as well as the impact of embedding dimensions on prediction accuracy.

3. Influence of the three model modules (especially the Koopman Operator configuration) on overall model performance.

Detailed modifications can be found in the revised version we submitted (mainly in Sec. 2, 3.1 and 5).

**For point #1: Section 2 & 3**

[Figure]

Revised Figure 6: (a) Calculation of the slip interval $\Delta T$. (b)-(d) slip intervals histogram and spectrogram of 3 experiments.

We apply ED to the learned Koopman operator (Eq. (4)) and obtain the decomposed complex eigenvalue (Table 2), which represents eigendynamic modes characterized by distinct amplitudes and periods. The results are in general agreement with the dominant period obtained from frequency spectral analysis of different slip experimental stresses (Figure 6b-d). ···

**For point #2: Section 4**

In addition, we find that the dimension of delayed embedding affects the prediction performance and prediction preference of HKAE to a larger extent. For example, in the slow-slip, low system dimension scenario, the overall prediction of HKAE with low embedding dimension is significantly better than that with high embedding dimension. For the fast-slow alternating slip and fast slip scenarios, the high-embedding dimension tends to obtain long-term stable prediction results, while the low-embedding dimension is able to obtain more accurate short-term prediction results (Figure 3 in the Supplementary Information). This suggests that the embedded dimension of HKAE need to be adjusted according to the slip activity state in practical applications.

**For point #3: Section 4**

To address these questions, informed by dynamic system theories, we pioneered a dynamic informed method, the HKAE, to predict the future shear stress of laboratory fault slips. The HKAE model is designed on the basis of delay embedding theory and Koopman theory and leverages the nonlinear fitting capabilities of neural networks and the systematic perspective of dynamic theories. The advantages of the HKAE include (1) multiscale modelling of laboratory slip systems under limited observations and (2) evolution mode and insights into laboratory slip from a dynamic systems perspective. The rationality of the HKAE architecture design was further verified in the ablation experiments of the three modules, especially the setup of the Koopman Operator module (Figure S6 in Supplementary Information).

2. **The authors tend to overstate their results, despite (a) clear instances of underestimation or overestimation in multiple cycles, and (b) performance that is comparable to other models, such as LSTM, in many cases.**

**[Response]:**

Thank you for your comments.

(1) For multiple cycles (i.e., intercyle in main text), we conclude that HKAE outperforms other methods, mainly from the goodness-of-fit of the slip intervals (i.e., Figure 10 a-2, b-2). Under statistical prediction with multiple cycles, HKAE is not numerically superior under all prediction leads. However, we believe that it is clearly **more meaningful to be able to predict precise slip intervals under multiple cycles of prediction**. In this case, HKAE clearly outperforms the other methods in that its prediction of the slip interval is more stable as the number of prediction steps increases, whereas the other methods show varying degrees of oscillation. In addition, HKAE also outperforms the other methods in terms of the statistics of the prediction results for the whole slip interval, as shown by the larger $R^2$ of $\Delta T$ although in the fast slip experiments, the $R^2$ of $\Delta T$ is negative, a phenomenon that occurs in all the prediction methods, and we have added a description of this limitation in the main text.

(2) For the performance of HKAE relative to other methods under different cases, we measure three main dimensions:

➢ First under the condition of intra-cycle 10s predicting 3s, Figure 8 shows the comparison of the prediction scores of different methods. Here we use the embedding dimension ($d = 100$) across experiments. Under the experiment of embedding dimension test (Figure S3) it can be found that better performance can be obtained under Exp. 4581 ($d = 60$) and Exp. 5198 ($d = 5$) in Figure 8. Secondly, under this experimental condition, we emphasize its prediction results in the stress-rise and stress-fall phases, and from the performance of the example represented by Figure 9, the prediction results of HKAE are superior, **especially in the stress-release phase**.

➢ Under the condition of inter-cycle 20s prediction for 10s, Figure 10 demonstrates the comparison of the prediction scores of different methods. Again, HKAE does not show a significant advantage in terms of statistical metrics (hence our use of the word comparable in the abstract), but it is able to predict slip intervals more accurately under these prediction conditions, which is certainly **a more meaningful characterization of the model's effectiveness**.

Based on your comments, we have made the following adjustments in the main text to make the content more objective.

For # Abstract

The HKAE performs dynamic modelling of laboratory fault systems and provides a continuous estimation of the future state of the system. It has been used in experiments with different slip behaviours and has the ability to predict shear stress variation during a slip cycle and slip activity during long-term seismic cycles. The HKAE outperforms traditional statistical methods while achieving results comparable to cutting-edge deep learning methods across multiple prediction scales. This is particularly evident in its accurate prediction of the stress release phase and precise estimation of the slip interval. More importantly, through dynamic theory and operator analysis in latent space, the HKAE provides insights into the stability of laboratory slip systems rather than full end-to-end black-box predictions.

For # Conclusion

In addition to the modelling performance, the analysis of the execution process of the HKAE can provide dynamic diagnostics for the laboratory slip system operating behind the shear stress observations, such as those of system trajectory behaviour, characteristic dynamic modes, and system stability. HKAE prediction results and dynamical system analyses show that slip behaviour, especially the long-term future prediction of fast-slip stress states, remains challenging.

3. **Additionally, the interpretability argument presented by the authors is a general observation about the system's dynamics rather than a concrete improvement in forecasting accuracy or model generalization.**

[Response]:

Thank you for your comments.

Interpretability is critical for applying deep learning in geosciences. Here, interpretability is defined relative to purely data-driven models: while time-series models like TCN and LSTM remain black-box, HKAE provides dynamical system-level insights, including dominant evolutionary modes and stability of system dynamics et al. We argue that such dynamical insights constitute a valid form of interpretability. Dynamical analysis of observational data is essential for understanding predictability and stability in Earth systems [1,2], particularly in fault slip scenarios where recent studies have adopted dynamical perspectives to reveal key behavioral patterns [3,4].

Unlike post-hoc explanation methods (e.g., SHAP, feature importance) that focus on variable correlations, HKAE's interpretability originates from first-principles dynamical theory. To prevent ambiguity, we explicitly defined this distinction in Section 2.3 of the revised manuscript.

**References:**

[1] Miller, S. A., Nur, A., & Olgaard, D. L. (1996). Earthquakes as a coupled shear stress-high pore pressure dynamical system. Geophysical Research Letters, 23(2), 197-200.

[2] Runge, J., Bathiany, S., Bollt, E., Camps-Valls, G., Coumou, D., Deyle, E., ... & Zscheischler, J. (2019). Inferring causation from time series in Earth system sciences. Nature communications, 10(1), 2553.

[3] Gualandi, A., Faranda, D., Marone, C., Cocco, M., & Mengaldo, G. (2023). Deterministic and stochastic chaos characterize laboratory earthquakes. Earth and Planetary Science Letters, 604, 117995.

[4] Gualandi, A., Dal Zilio, L., Faranda, D., & Mengaldo, G. (2024). Similarities and differences between natural and simulated slow earthquakes. Geophysical Research Letters, 51(14), e2024GL109845.

4. **The manuscript lacks a discussion on the model's applicability to other parameters, such as time to failure, as well as its relevance to real-world field studies. It is evident that stress values are only accessible in laboratory experiments and cannot be directly extrapolated to tectonic earthquakes. Therefore, what are the potential applications of this model in such scenarios?**

**[Response]:**

Thank you for your comments.

(1) Our decision to exclude TTF prediction stems from two considerations:

   ➢ Existing works on laboratory earthquake forecasting predominantly predict both TTF and stress states, with high consistency between the two predictions [1-3].

   ➢ As a dynamics-inspired method, HKAE prioritizes direct system state variables (shear stress) over engineered features like TTF. Shear stress explicitly characterizes the system's physical state, whereas TTF represents a derived phenomenological metric.

(2) While stress measurements are inaccessible in tectonic settings, recent studies demonstrate that GNSS or seismogram time series can serve as proxies for stress state variations [4-6], analogous to acoustic emission-stress correlations in laboratory earthquakes. Neural network architecture of HKAE could adapt to field data by modifying input-output mappings structures (e.g., add a network branch to learn the relationship between stress and acoustic emissions). In particular, recent work has shown that machine learning algorithms can detect states such as displacement of active faults directly from continuous waveforms [6].

We have added these ideas in Discussion. In our next work, we will consider the prediction of actual measurable metrics such as TTF on the one hand, and test the prediction of accessible data from tectonic seismic on the other.

**References:**

[1] Rouet-Leduc, B., Hulbert, C., Lubbers, N., Barros, K., Humphreys, C. J., & Johnson, P. A. (2017). Machine learning predicts laboratory earthquakes. Geophysical Research Letters, 44(18), 9276-9282.

[2] Shokouhi, P., Girkar, V., Rivière, J., Shreedharan, S., Marone, C., Giles, C. L., & Kifer, D. (2021). Deep learning can predict laboratory quakes from active source seismic data. Geophysical Research Letters, 48(12), e2021GL093187.

[3] Wang, C., Xia, K., Yao, W., & Marone, C. (2025). Generalizable deep learning models for predicting laboratory earthquakes. Communications Earth & Environment, 6(1), 219.

[4] Rouet-Leduc, B., Hulbert, C., & Johnson, P. A. (2019). Continuous chatter of the Cascadia subduction zone revealed by machine learning. Nature Geoscience, 12(1), 75-79.

[5] Johnson, C. W., & Johnson, P. A. (2024). Seismic features predict ground motions during repeating caldera collapse sequence. Geophysical Research Letters, 51(11), e2024GL108288.

[6] Johnson, C. W., Wang, K., & Johnson, P. A. (2025). Automatic speech recognition predicts contemporaneous earthquake fault displacement. Nature Communications, 16(1), 1069.

We suggest that the HKAE can achieve competitive modelling of seismic activity and diagnose the dynamic behaviour of regional seismic systems by incorporating dynamic system theory. … In addition, stress is not directly accessible under tectonic seismic environmental conditions, which increases the difficulty of applying HKAE under real conditions. However, recent studies have shown that time-series observations, such as GNSS and seismometers, exhibit the feasibility of serving as a proxy for the state change of stress in tectonic earthquakes, and this relationship is similar to that between acoustic emission signals and stress changes in laboratory earthquakes (Johnson et al., 2024; 2025). HKAE has advantages for data fusion due to its flexible neural network architecture implementation.  Therefore, the generalizability of the model can be improved by integrating external data such as historical acoustic emissions or other measurable laboratory observations by means such as adding coding branches.

**5.  Furthermore, while shear stress exhibits a highly periodic behavior in double-shear tests, rough fault stick-slip experiments (Goebel et al., 2012; Dresen et al., 2020) have demonstrated that roughness evolution significantly influences stress cycles. The authors should include a discussion on this aspect.**

**[Response]:**
We sincerely appreciate the reviewer's valuable comments on the evolution of complex stress changes. In response, we have strengthened the Discussion section (Section 5) to explicitly address the implications of heterogeneous fault conditions (e.g., roughness effects) on stress predictability.

This work intentionally focuses on three fundamental slip regimes (fast, slow, and alternating fast-slow slips) to establish baseline dynamics under bi-shear experiments. These regimes provide essential benchmarks for evaluating HKAE's core capabilities (Section 3). We fully agree that fault roughness amplifies nonlinear stress variations, as evidenced by prior studies (Goebel et al., 2012; Dresen et al., 2020). Such complexity warrants dedicated investigation beyond the current triaxial shear experimental framework. We added related points in Discussion.

We suggest that the HKAE can achieve competitive modelling of seismic activity and diagnose the dynamic behaviour of regional seismic systems by incorporating dynamic system theory. Currently there are two main challenges in the application of HKAE to actual tectonic conditions. One is that the stress state changes of actual tectonic earthquakes may be complex. In order to verify the modelling capability of HKAE, we tested it using a typical double-shear experiment representing slip fast and slow with alternation. It is shown that the stress changes are more complex under rough fault viscous slip experimental conditions (Dresen et al., 2020). Although recent studies have shown that slip Time-To-Failures (TTFs) under high roughness can be predicted using machine learning (Karimpouli et al., 2023), slip dynamical system properties under high roughness remain currently undiscussed, which may affect the future predictive performance of HKAE under such more complex conditions of slip. …

6.  **Figures 8 and 10 contain numerous plots. For instance, in Figure 10, if the first two columns represent different views of the same plot, they should be visually linked using lines or another connecting element. And use some kind of numbering gor them.**

**[Response]:**

Thank you for your comments.

We have adjusted the presentation of Figure 8, 9 and 10 in the revised version. We add the numbers of panels to quickly localization.

[Figure]

**Revised Figure 8: Genenral evaluation for 3 s lead prediction using historical 10 s shear stress, with R² and RMSE used as evaluation metrics. (a)-(c) for Exp. 4581, Exp. 4679 and Exp. 5198 respectively.**

[Figure]

**Revised Figure 9: 3 s leading prediction details during different phases of stress variation for different laboratory datasets. (a), (c), and (e) show the HKAE results, whereas (b), (d), and (f) show the LSTM results. The left panels (x-1) present the total 3 s leading predictions for the test set. The right panels (x-2, x-3) illustrate the predictions in the 3 s horizon, with R2 and RMSE used as evaluation metrics. Predictions with higher metrics are highlighted in red.**

**(a) 10 s Lead Time Prediction Using Historical 20 s Shear Stress for Exp. 4581**

**(a-1) Statistical evaluation metrics of lead predictions**

**(a-2) Slip intervals evaluation of lead predictions**

**(b) 10 s Lead Time Prediction Using Historical 20 s Shear Stress for Exp. 4679**

**(b-1) Statistical evaluation metrics of lead predictions**

**(b-2) Slip intervals evaluation of lead predictions**

Revised Figure 10: General evaluation for 10 s lead prediction using historical 20 s shear stress, with $R^2$, RMSE and $R^2$ of event intervals (Eq. 18) used as evaluation metrics. (a)-(b) for Exp. 4581, Exp. 4679 respectively. (a-1) and (b-1) illustrate the metrics variation with prediction leads. (a-2) and (b-2). The prediction results of the sliding prediction process for the sliding intervals were counted and compared with the real sliding intervals (Figure 6b-d).

7. **Figure 10: What does a negative R2 value indicate?**

**[Response]:**

We thank the reviewer for raising this important point.

A negative $R^2$ indicates that the prediction results of MORNING are worse than the mean value of the observed data used directly. The negative $R^2$ in Fig. 10 evaluates the results of the $\Delta T$ prediction of the slip period for the HKAE and LSTM for the 10s lead case. From the figure, it can be observed that in the fast slip experiment (Exp. 4581), the slip period occurs in segments of 8-12 s. The negative $R^2$ values for both HKAE and LSTM suggest that neither of the currently employed methods may be able to capture the evolutionary trend of the fast slip system, although the negative $R^2$ values for HKAE are slightly smaller than those for LSTM.

We add the reasons for the occurrence of negative $R^2$ values in Sec. 3.3.

To further assess the ability of the prediction method to model the event cycle, we counted the predictions for the slip intervals in the prediction window in the sliding prediction experiment, and assessed the goodness of fit between the predictions and the true intervals (Figure 10a-2, b-2). Considering the event cycle predictions over the entire prediction window, the HKAE also has a advantage over the LSTM, as demonstrated by the fact that its cycle predictions are closer to the identity line. However, the $R^2$ scores of slip intervals are negative in Exp.4581, which indicates that both HKAE and LSTM have limited ability to capture the evolutionary features of fast slip systems.